# AhR and SHP regulate phosphatidylcholine and S-adenosylmethionine levels in the one-carbon cycle

Young-Chae Kim[1], Sunmi Seok[1], Sangwon Byun[1], Bo Kong [2], Yang Zhang[3], Grace Guo[2], Wen Xie[4], Jian Ma[3], Byron Kemper[1] & Jongsook Kim Kemper[1]

Phosphatidylcholines (PC) and S-adenosylmethionine (SAM) are critical determinants of hepatic lipid levels, but how their levels are regulated is unclear. Here, we show that *Pemt and Gnmt*, key one-carbon cycle genes regulating PC/SAM levels, are downregulated after feeding, leading to decreased PC and increased SAM levels, but these effects are blunted in small heterodimer partner (SHP)-null or FGF15-null mice. Further, aryl hydrocarbon receptor (AhR) is translocated into the nucleus by insulin/PKB signaling in the early fed state and induces *Pemt and Gnmt* expression. This induction is blocked by FGF15 signaling-activated SHP in the late fed state. Adenoviral-mediated expression of AhR in obese mice increases PC levels and exacerbates steatosis, effects that are blunted by SHP co-expression or *Pemt* downregulation. PEMT, AHR, and PC levels are elevated in simple steatosis patients, but PC levels are robustly reduced in steatohepatitis-fibrosis patients. This study identifies AhR and SHP as new physiological regulators of PC/SAM levels.

[1] Department of Molecular and Integrative Physiology, University of Illinois at Urbana-Champaign, Urbana, IL 61801, USA. [2] Department of Pharmacology and Toxicology, School of Pharmacy, Rutgers University, Piscataway, NJ 08854, USA. [3] Department of Computational and Comparative Genomics Lab, School of Computer Science, Carnegie Melon University, Pittsburgh, PA 15213, USA. [4] Department of Pharmaceutical Sciences, School of Pharmacy, University of Pittsburgh, Pittsburgh, PA 15261, USA. Correspondence and requests for materials should be addressed to Y.-C.K. (email: ychaekim@illinois.edu) or to J.K.K. (email: jongsook@illinois.edu)

Nonalcoholic fatty liver disease (NAFLD) is the most common cause of chronic liver diseases and is frequently associated with diabetes, dyslipidemia, and cardiovascular diseases[1]. NAFLD can progress to steatohepatitis (NASH) and further to fatal liver diseases, such as fibrosis, cirrhosis, and liver cancer[2]. The hallmark of NAFLD is abnormal accumulation of triglycerides (TGs) in the liver, but the underlying mechanisms for the development of NAFLD are poorly understood. Emerging evidence indicates that methyl metabolites in the one-carbon (1C) cycle, such as phosphatidylcholines (PC) and S-adenosylmethionine (SAM), are critical determinants of hepatic TG levels[3–5]. Two enzymes that have key roles in the synthesis of PC from SAM and phosphatidylethanolamine (PE) are: phosphatidylethanolamine N-methyltransferase (PEMT), which catalyzes methyl transfer from SAM to PE to form PC and glycine N-methyltransferase (GNMT), which catalyzes the conversion of SAM to SAH as illustrated in Fig. 1a. A balance in the levels of PC and SAM is critical for hepatic lipid metabolism, and this balance is disrupted in NAFLD[6], but the transcriptional mechanisms regulating the 1C cycle are not completely understood.

Aryl hydrocarbon receptor (AhR) is a ligand-regulated transcription factor that senses environmental toxicants, such as 2,3,7,8-Tetrachlorodibenzo-p-dioxin (TCDD), commonly called dioxin[7, 8]. Upon ligand binding, AhR is translocated into the nucleus, where AhR forms a heterodimer with AhR nuclear translocator (Arnt) and binds to dioxin response elements to induce transcription of genes involved in xenobiotic metabolism, including Cyp1a1[7]. In addition to its xenobiotic function, AhR also has endobiotic physiological functions in cell proliferation, differentiation, immune suppression, and male infertility[9–11]. AhR was recently also shown to play an important role in hepatic lipid metabolism[12]. AhR increases hepatic TG levels by increasing expression of CD36, a key transporter for fatty acid uptake[12–14]. The possibility that AhR may increase liver TG levels and promote steatosis in part through regulation of the 1C cycle genes has not been reported.

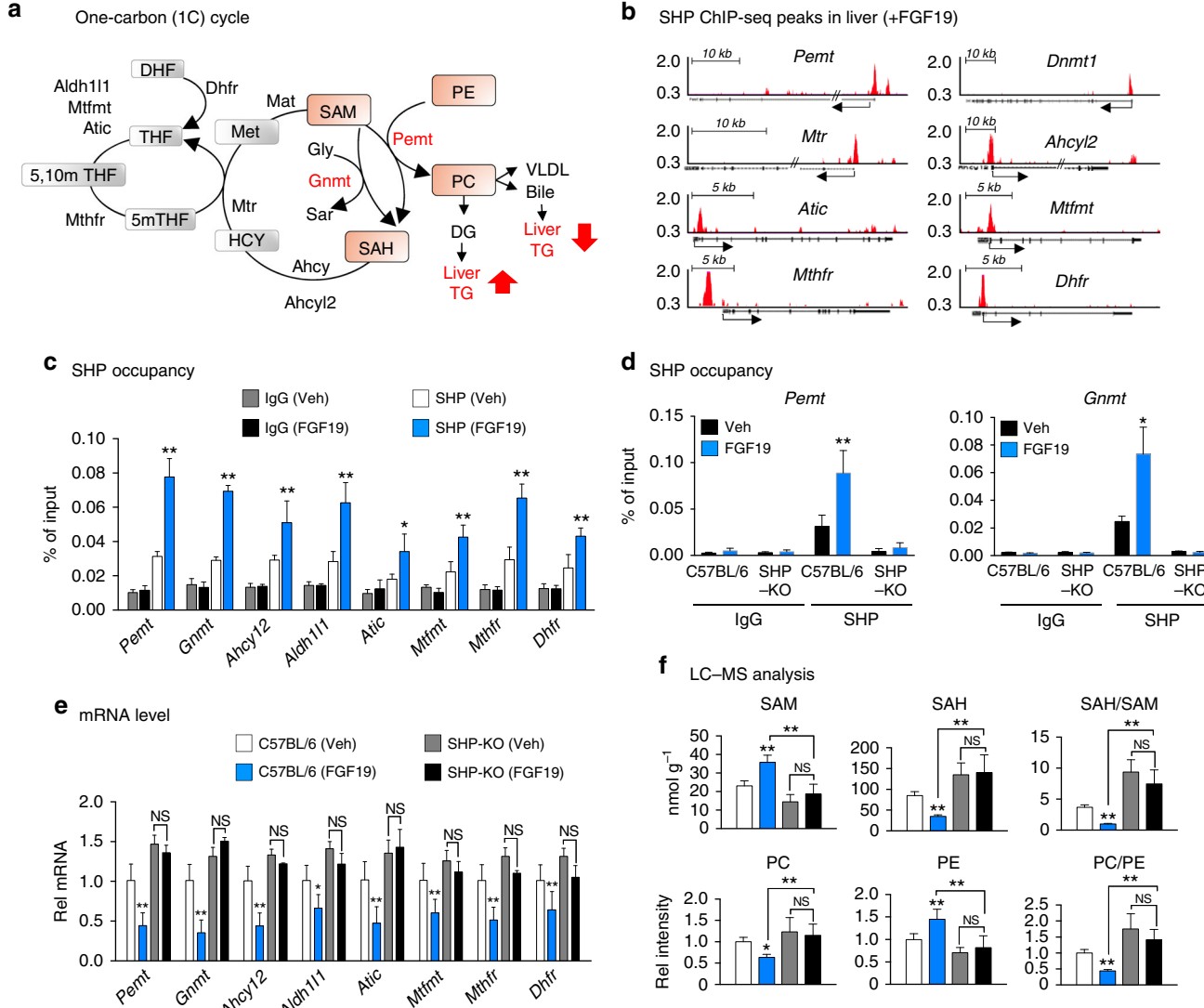

**Fig. 1** FGF19 inhibition of hepatic PC production is SHP dependent. **a** Schematic diagram of the 1C cycle. **b** From liver ChIP-seq data for mice treated with FGF19[26], positions of SHP binding peaks at the promoter regions of key genes involved in the 1C cycle are shown. **c**, **d** Effects of FGF19 treatment for 2 h on SHP occupancy determined by ChIP at 1C genes in C57BL/6 mice (**c**), and comparison of occupancy of SHP at Pemt and Gnmt genes in C57BL/6 and SHP-KO mice (**d**). **e**, **f** C57BL/6 or SHP-KO mice were treated with FGF19 for 6 h, and mRNA levels of 1C cycle genes (**e**) or 1C cycle metabolites levels (**f**) were determined by qRT-PCR or LC–MS, respectively. **c**–**f** Means ± SD are shown ($n = 5$ mice), and statistical significance was measured using the two-way ANOVA with the FDR post-test and *$P < 0.05$, **$P < 0.01$, NS, not statistically significant

Small heterodimer partner (SHP, NR0B2) is an unusual orphan nuclear receptor that lacks a DNA binding domain[15]. SHP acts as a transcriptional co-repressor and represses the activity of many transcription factors by recruiting repressive histone-modifying proteins, such as G9a histone methyltransferase and LSD1 histone demethylase[16, 17]. SHP is transcriptionally induced by the bile acid nuclear receptor FXR (farnesoid X receptor) after feeding, but SHP protein stability and its gene repression activity are also increased post-translationally in response to FGF15/19 (hFGF19, mFGF15) signaling in the late fed state[18–20]. In addition to its well-known function of repression of bile acid production[21], SHP regulates lipid metabolism and inflammation[22, 23]. SHP has also been shown to mediate circadian regulation of TG metabolism and homocysteine levels[24, 25]. Recent global liver chromatin immunoprecipitation sequencing (ChIP-seq) analysis has identified new functions for SHP in repression of hepatic autophagy and cholesterol biosynthesis in the late fed state[26, 27].

In this study, we show a novel function of AhR and SHP in the regulation of hepatic PC/SAM levels through transcriptional control of 1C genes. We present evidence that AhR is translocated into the nucleus by insulin/protein kinase B (PKB) signaling early after feeding and induces transcription of *Pemt and Gnmt* and this transcriptional induction is antagonized in the late fed state by fibroblast growth factor 15/19 (FGF15/19) signal-activated SHP. We show that adenoviral-mediated expression of AhR exacerbates hepatic TG accumulation in dietary obese mice, which is blocked by expression of SHP or downregulation of Pemt. We further show the human relevance of our findings utilizing liver samples of NAFLD patients.

## Results

**FGF19-mediated inhibition of PC production requires SHP.** From published liver ChIP-seq data from mice treated with the late fed-state hormone FGF19[26], SHP binding peaks were detected at the promoter regions of numerous genes involved in the 1C cycle (Fig. 1b). Confirming these results, in standard mouse liver ChIP assays, treatment with a pharmacological dose of FGF19 increased binding of SHP at all tested 1C cycle genes, including *Pemt and Gnmt*, two key genes in regulating PC and SAM levels (Fig. 1c), while the binding of SHP at *Pemt and Gnmt* was not detected in SHP-knockout (SHP-KO) mice (Fig. 1d). Further, mRNA levels of the 1C cycle genes were decreased by FGF19 treatment in control C57BL/6 mice, but not in SHP-KO mice (Fig. 1e).

Consistent with these transcriptional changes, FGF19 treatment increased hepatic SAM and PE levels and decreased SAH and PC levels in control C57BL/6 mice, so that the PC/PE and SAH/SAM ratios were decreased, but these changes were blunted in SHP-KO mice (Fig. 1f). These results demonstrate that FGF19 inhibits the expression of 1C cycle genes and decreases hepatic PC/PE and SAH/SAM ratios in the liver in a SHP-dependent manner.

**SHP and FGF15/19 physiologically regulate PC/SAM levels.** FGF15/19 is a late-fed-state hormone that is induced by feeding-sensing bile acid nuclear receptor FXR[28–30] and mediates postprandial responses independent of insulin action[31]. To assess the physiological relevance of SHP-dependent FGF15/19 inhibition of PC levels (Fig. 1), we examined feeding-mediated changes in hepatic expression of 1C cycle genes and key methyl metabolite levels in SHP-KO or FGF15-KO mouse models.

Feeding led to decreased mRNA levels of all the 1C genes tested (Fig. 2a), and decreased PC/PE and SAH/SAM ratios (Fig. 2b), and importantly, these feeding-mediated effects were reversed or blunted in SHP-KO mice. Furthermore, the feeding-mediated effects on mRNA levels of 1C genes (Fig. 2c) and the changes in

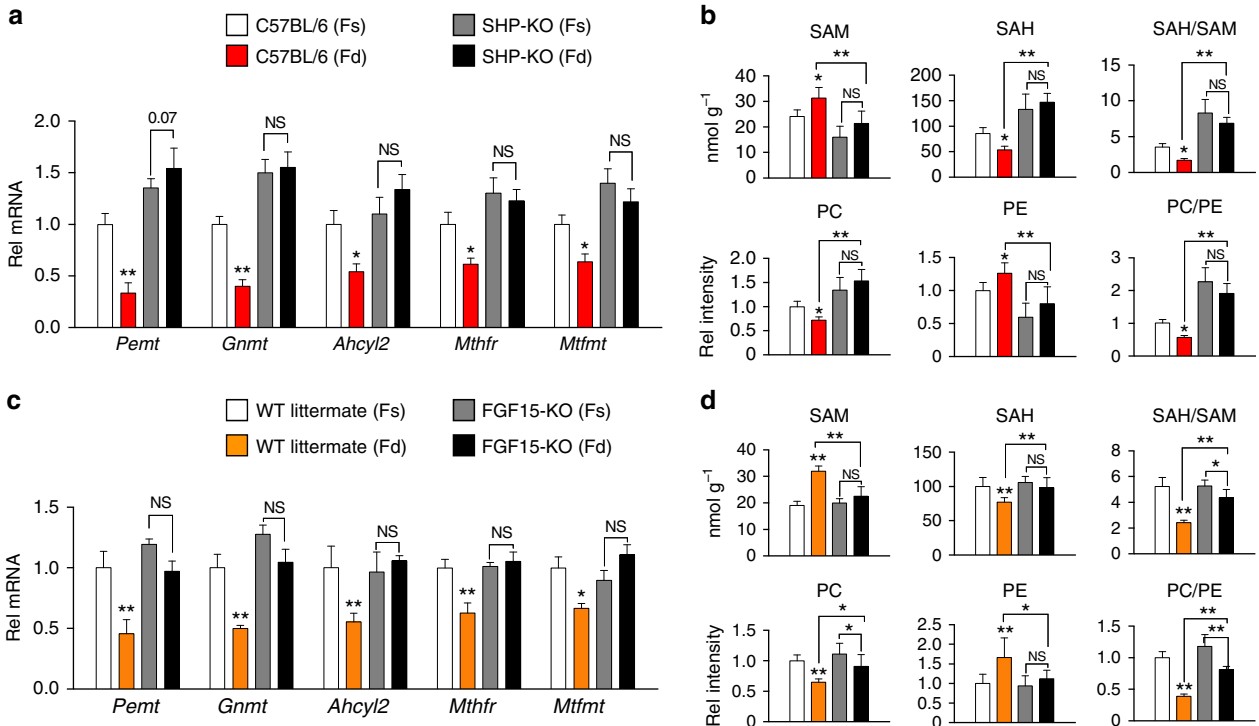

**Fig. 2** FGF15 and SHP are physiological repressors of PC production in the fed state. C57BL/6 mice and SHP-KO mice (**a**, **b**) or FGF15-KO and WT littermate (**c**, **d**) mice were fasted for 12 h (Fs) or fasted and refed for 6 h (Fd). Hepatic mRNA levels of selected 1C cycle genes (**a**, **c**) and 1C cycle metabolites (**b**, **d**) were measured by qRT-PCR and LC–MS analysis, respectively. Means ± SD are shown (n = 5 mice), and statistical significance was measured using the two-way ANOVA with the FDR post-test and *P < 0.05, **P < 0.01, NS, not statistically significant

the PC/PE and SAH/SAM ratios (Fig. 2d) were also attenuated in FGF15-KO mice compared to control wild-type (WT) littermates. Consistent with these findings, activation of FXR by treatment with its agonist, GW4064, which induces expression of both *Shp* and *Fgf15*, led to decreased mRNA levels of key 1C genes, including *Pemt* and *Gnmt*, and decreased ratios of PC/PE and SAH/SAM levels in control C57BL/6 mice, but these effects were attenuated in SHP-KO mice (Supplementary Figure 1). These findings demonstrate that SHP and FGF15 are physiological regulators of hepatic PC and SAM levels in the fed state.

**AhR activates 1C genes and increases hepatic PC levels**. We next investigated the mechanism by which SHP represses expression of 1C cycle genes. Since SHP does not directly bind to

DNA[15], we identified binding sites of potential transcription factors that might recruit SHP to 1C genes. AhR binding motifs, TGCGTG, were present most frequently within SHP binding peak regions in 1C genes, including *Pemt*, *Gnmt*, *Ahcyl2*, *Mtr*, and *Mthfr*, although binding sites for other factors, such as SREBP-1 and CREB, were also detected in some of these genes (Fig. 3a, Supplementary Table 3). In mouse liver ChIP assays, occupancy of AhR at the SHP binding peak regions of these genes was significantly increased after feeding (Fig. 3b), which suggests a potential role for an AhR–SHP axis in transcriptional regulation of 1C genes in the fed state.

To examine whether AhR can transactivate 1C genes, we performed cell-based reporter assays using *Pemt*- and *Gnmt*-luciferase reporter constructs which contain the SHP binding peak region in *Pemt* and *Gnmt*. Expression of constitutively active

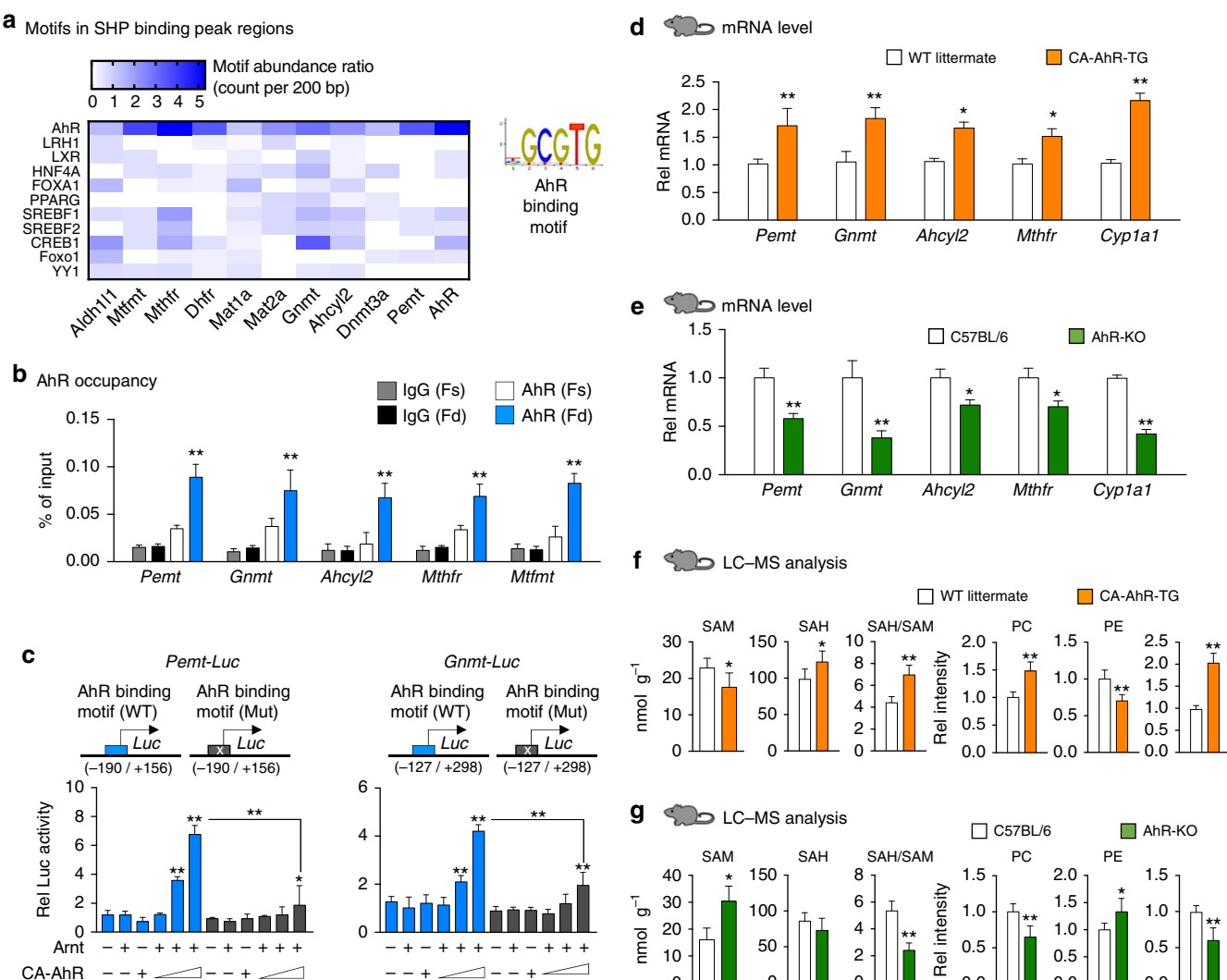

**Fig. 3** AhR transactivates 1C cycle genes and increases hepatic PC levels. **a** Potential binding sites for transcription factors within SHP peak regions at 1C genes were identified using the JASPAR database, and the abundance of representative factor binding sites is shown (left). An AhR/Arnt binding motif is shown (right). **b** Mice (n = 5 mice per group) were fasted overnight (Fs) or fasted and refed for 6 h (Fd), and AhR occupancy within the SHP binding peak regions at the 1C genes was determined by liver ChIP assays. **c** Hepa1c1c7 cells were transfected with a *Pemt-luc* or *Gnmt-luc* construct containing the WT or mutated AhR binding site in the *Pemt* promoter along with expression plasmids as indicated and treated with FGF19 for 2 h, and luciferase activity was measured (n = 5). **d–g** AhR transgenic mouse studies: Liver tissues were collected from WT littermate and CA-AhR transgenic mice (n = 4–7 mice per group), or C57BL/6 (WT control) and AhR-KO mice (n = 5–10 mice per group) and **d**, **e** mRNA levels of selected 1C genes and **f**, **g** 1C cycle metabolites were measured by qRT-PCR and LC–MS analysis, respectively. Means ± SD are shown, and statistical significance was measured using **d–g** Student's *t*-test or **b**, **c** two-way ANOVA with the FDR post-test. *P < 0.05, **P < 0.01, NS, not statistically significant

(CA)-AhR with Arnt increased transactivation of these promoters and the increase was largely blocked by mutation of the AhR binding motif (Fig. 3c). In addition, treatment of cells expressing AhR and Arnt, with an AhR agonist, 3-methylcholanthrene, increased transactivation of the *Pemt* promoter by AhR (Supplementary Figure 2). Taken together with the ChIP studies (Fig. 3b), these results suggest that AhR can bind to the *Pemt* promoter and transactivate the gene.

The induction of *Pemt* by AhR suggests that AhR may regulate PC and SAM levels in vivo by its effects on the expression of 1C cycle genes. To determine the in vivo functional role of AhR in the expression of 1C genes and PC metabolism, we utilized two genetic AhR mouse models, CA-AhR transgenic and AhR-KO mice[12, 14, 32]. Compared to control mice, hepatic mRNA levels of *Pemt*, *Gnmt*, *Ahcyl2*, and *Mthfr*, as well as a well-known AhR target gene, *Cyp1a1*, were increased in CA-AhR-transgenic mice (Fig. 3d), but were significantly decreased in AhR-KO mice (Fig. 3e).

Consistent with the changes in 1C gene expression, the ratios of PC/PE and SAH/SAM were significantly increased and decreased in the CA-AhR and AhR-KO mice, respectively (Fig. 3f, g). These results demonstrate that AhR is a novel regulator of hepatic PC and SAM levels through transcriptional activation of 1C cycle genes.

**Insulin/PKB signaling mediates nuclear translocation of AhR.** AhR is translocated into the nucleus upon exposure to its xenobiotic ligands, such as TCDD[7, 8]. To determine the effect of feeding on nuclear localization of AhR, we examined the effect of fed-state hormones, insulin and FGF19, and bile acids on nuclear levels of AhR in mouse Hepa1c1c7 cells. AhR was detected mostly in the cytoplasm in cells grown in serum-free low-glucose medium, while in complete medium AhR was mainly in the nucleus of Hepa1c1c7 (Fig. 4a, Supplementary Figure 12a) or human HepG2 (Supplementary Figures 3a and 13a) cells and ARNT, DNA binding partner of AhR, was detected mostly in the nucleus (Fig. 4a, Supplementary Figures 3a, 12a, and 13a). Treatment of these cells in the low-glucose medium with insulin led to nuclear localization of AhR, whereas treatment with a primary bile acid, chenodeoxycholic acid (CDCA), or FGF19 did not (Fig. 4a, Supplementary Figures 3a, 12a, and 13a). Insulin activates the PI3K/PKB-AKT and Ras/MAPK-ERK pathways[33]. Co-treatment of insulin with PKB124005, a PKB inhibitor blocked the insulin-mediated nuclear localization of AhR while a MEK inhibitor, PD98059, only modestly reduced nuclear levels of AhR (Fig. 4b, Supplementary Figures 3b, 12a, and 13a), which is consistent with a previous report that PD98059 is an AhR antagonist[34]. These nuclear localization studies suggest that AhR is translocated into the nucleus in response to insulin/PKB signaling.

We next confirmed whether the translocation of AhR by insulin affects expression of *Pemt*. We observed that co-occupancy of AhR and ARNT at the *Pemt* promoter was increased (Supplementary Figure 4) and the levels of pre-mRNA, indicative of transcriptional induction of *Pemt*, as well as the known target of AhR, *Cyp1a1*, were increased by 45 min after treatment of insulin (Fig. 4c), suggesting that insulin-mediated AhR translocation into nucleus regulates transcriptional activation of *Pemt*. While insulin did not increase nuclear levels of SHP, FGF19 increased nuclear SHP levels, which is consistent with previous studies[20, 26]. These results, together, indicate that nuclear AhR levels are increased by an early fed-state hormone, insulin, via the PKB signaling pathway, and nuclear SHP levels are increased by a late fed-state hormone, FGF15/19.

**SHP interacts with AhR and inhibits AhR transactivation.** Since AhR binding motifs were present within SHP peak regions

at all 1C cycle genes analyzed (Fig. 3a) and nuclear localization of both AhR and SHP is increased after addition of complete medium (Fig. 4a, b, Supplementary Figure 12a), we next examined whether AhR and SHP functionally interact with each other in the regulation of 1C genes. SHP was previously shown to directly interact with Arnt, but not with AhR[35]. In co-immunoprecipitation (CoIP) studies using mouse liver whole cell extracts, feeding or FGF19 treatment increased the level of SHP in AhR immunoprecipitates (Fig. 4d). In addition, feeding increased nuclear localization of both AhR and SHP in mouse liver as detected by immunofluorescence (IF; Fig. 4e), which is consistent with increased nuclear levels of AhR in Hepa1c1c7 cells after exposure to complete medium (Fig. 4a, Supplementary Figure 12a).

We next examined whether AhR, ARNT, and SHP co-occupy the promoter of *Pemt* in a chromatin context. In re-ChIP assays in mouse liver, feeding increased occupancy of SHP and ARNT in AhR-bound chromatin from the *Pemt* promoter (Fig. 4f) indicating co-occupancy of the promoter by AhR, ARNT, and SHP. To determine whether the interaction between AhR/ARNT and SHP is functional, the effects of SHP on the transactivation of the *Pemt and Gnmt* promotors by AhR/ARNT (Fig. 3c) were examined. Exogenous expression of SHP inhibited the increase in *Pemt* or *Gnmt* promoter activity mediated by expression of CA-AhR and Arnt (Fig. 4g, Supplementary Figure 5). In addition, expression of LRH-1, a known transcriptional activator of 1C genes[36], also increased the activity of these promoters and the increase was blocked by expression of SHP (Supplementary Figure 5). These results indicate that feeding increases the association of SHP with the AhR complex and co-occupancy at the *Pemt* promoter and that the interaction is transcriptionally functional.

**Temporal transcriptional regulation of *Pemt* by AhR and SHP.** FGF15/19 reaches peak serum levels about 3 h after feeding, while insulin levels peak considerably earlier at about 30 min[29, 37]. Since AhR nuclear levels are increased by insulin and SHP nuclear levels are increased by FGF19 (Fig. 4b, Supplementary Figure 12a), we next examined whether AhR is recruited to the *Pemt* promoter early after feeding and SHP is recruited later. Interestingly, in liver ChIP assays, binding of both AhR and ARNT to the *Pemt* promoter was detected as early as 1 h after feeding, while SHP binding was not detected at 1 h, but was increased by 2 h with maximal binding at 4 h (Fig. 4h). Pre-mRNA levels of *Pemt* were increased at 1 to 2 h after feeding, but were decreased by 4 h, which is consistent with early insulin-mediated binding of the activator AhR followed by later FGF19-mediated binding of the co-repressor SHP (Fig. 4i). These results, taken together with the subcellular localization studies (Fig. 4a, b, d, e, Supplementary Figure 12a), indicate that AhR is translocated into the nucleus and activates transcription of *Pemt* early after feeding, while SHP is translocated into the nucleus and recruited to AhR-bound chromatin at *Pemt* relatively late after feeding and inhibits the AhR-mediated transactivation of this gene.

**SHP inhibits AhR induction of its own gene expression.** From published liver ChIP-seq data from mice[26], a SHP binding peak was detected at the promoter of *AhR* (Fig. 5a, top). Interestingly, this region also contains multiple AhR binding motifs (Fig. 5a, bottom). We, thus, hypothesized that AhR might induce its own gene expression and further that SHP may inhibit the AhR-mediated induction.

To test this idea, we first examined if occupancy of SHP is increased at AhR-bound chromatin of the *AhR* gene promoter. In liver re-ChIP assays, SHP occupancy at the *AhR* promoter was

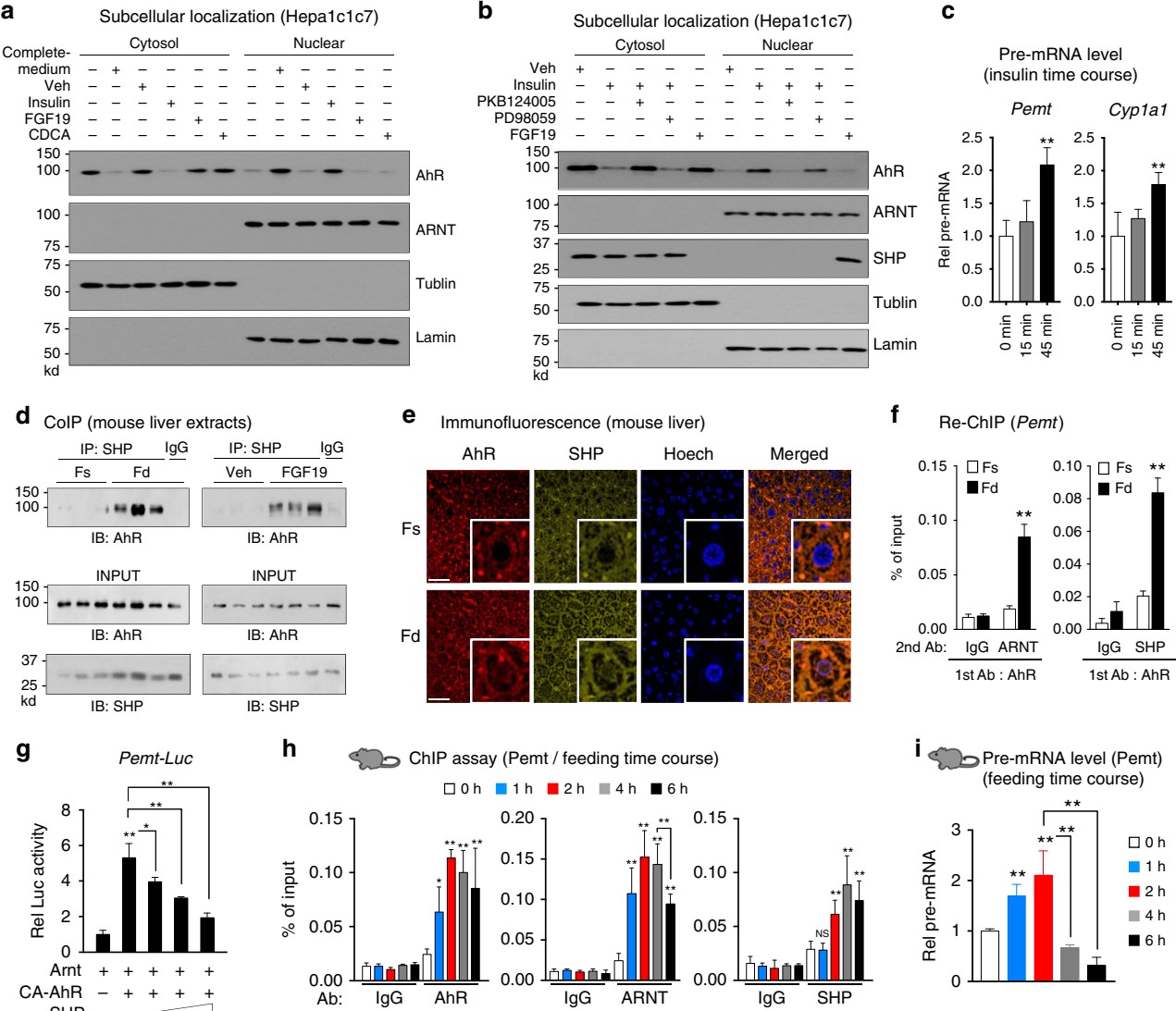

**Fig. 4** SHP inhibits AhR transactivation of *Pemt* in response to feeding or FGF19 treatment. **a, b** Hepa1c1c7 cells were grown in low-glucose and serum-free media for 12 h, and transferred for 15 min to complete medium or treated for 15 min with insulin, FGF19, or CDCA, and then cells were harvested. Cells were pre-treated with a PKB inhibitor, PKB124005, or an ERK inhibitor, PD98059, for 30 min prior to insulin treatment as indicated. Levels of AhR and SHP in the cytoplasmic and nuclear fractions were determined by IB. Consistent results were observed from two independent studies. Full size immunoblots are in Supplementary Figure 12a. **c** Hepa1c1c7 cells were treated with insulin for 15 min or 45 min, and pre-mRNA levels of *Pemt* and *Cyp1a1* were measured (*n* = 6). **d** Mice were fasted for 12 h (Fs) or fasted and refed for 4 h (Fd) or fasted 4 h and treated with FGF19 for 2 h, liver nuclear extracts were prepared, and the interaction of SHP and AhR was examined by CoIP. **e, f** Mice were fasted overnight (Fs) or fasted and refed for 6 h (Fd). **e** Cellular localization of AhR and SHP in mouse liver sections was examined by IF. Scale bar: 100 μM. **f** Liver chromatin was immunoprecipitated with AhR antibody, then eluted, and re-precipitated with ARNT or SHP antibody to examine the occupancy of SHP at AhR-bound chromatin at the *Pemt* promoter (*n* = 5 mice). **g** Hepa1c1c7 cells were transfected with the indicated plasmids, and then treated with FGF19 for 2 h and luciferase activity was determined (*n* = 5). **h, i** Mice (*n* = 5 mice per group) were fasted for 12 h and then refed for the indicated times, and the occupancy of AhR, ARNT, and SHP at the *Pemt* promoter (**h**) or *Pemt* pre-mRNA levels (**i**) were determined by liver ChIP assay or by qRT-PCR, respectively. Means ± SD are shown, and statistical significance was measured using the **c**, **g**, **i** one- or **f**, **h** two-way ANOVA with the FDR post-test. *$P < 0.05$, **$P < 0.01$, NS, statistically not significant

markedly increased in AhR-bound chromatin in mice after feeding (Fig. 5b). In cell-based reporter assays, expression of AhR and ARNT increased transactivation of the AhR-luc activity and the increase was blocked by expression of SHP or mutation of the AhR binding motif (Fig. 5c, Supplementary Figure 6). These results suggest that AhR can induce its own expression and the induction is likely inhibited by SHP.

To determine whether *AhR* gene expression is increased early after a meal and inhibited in the late fed state, *AhR* gene expression was examined as a function of time after feeding. Pre-mRNA levels of *AhR* were transiently increased up to 2 h after

feeding, but then rapidly decreased by 4 to 6 h to levels less than those in fasted mice (Fig. 5d). These results are consistent with initial transcriptional activation of *AhR* in early fed state, followed by inhibition by SHP in the late fed state, as was also observed for *Pemt* (Fig. 4g–i).

To assess the role of SHP inhibition of *AhR* expression in vivo, we analyzed the effect of feeding on hepatic expression of AhR in control C57BL/6 and SHP-KO mice in the late fed state. Consistent with the decreased AhR pre-mRNA levels (Fig. 5d), at 6 h after feeding, AhR mRNA levels decreased in WT mice, while in SHP-KO mice levels were about 60% higher than in

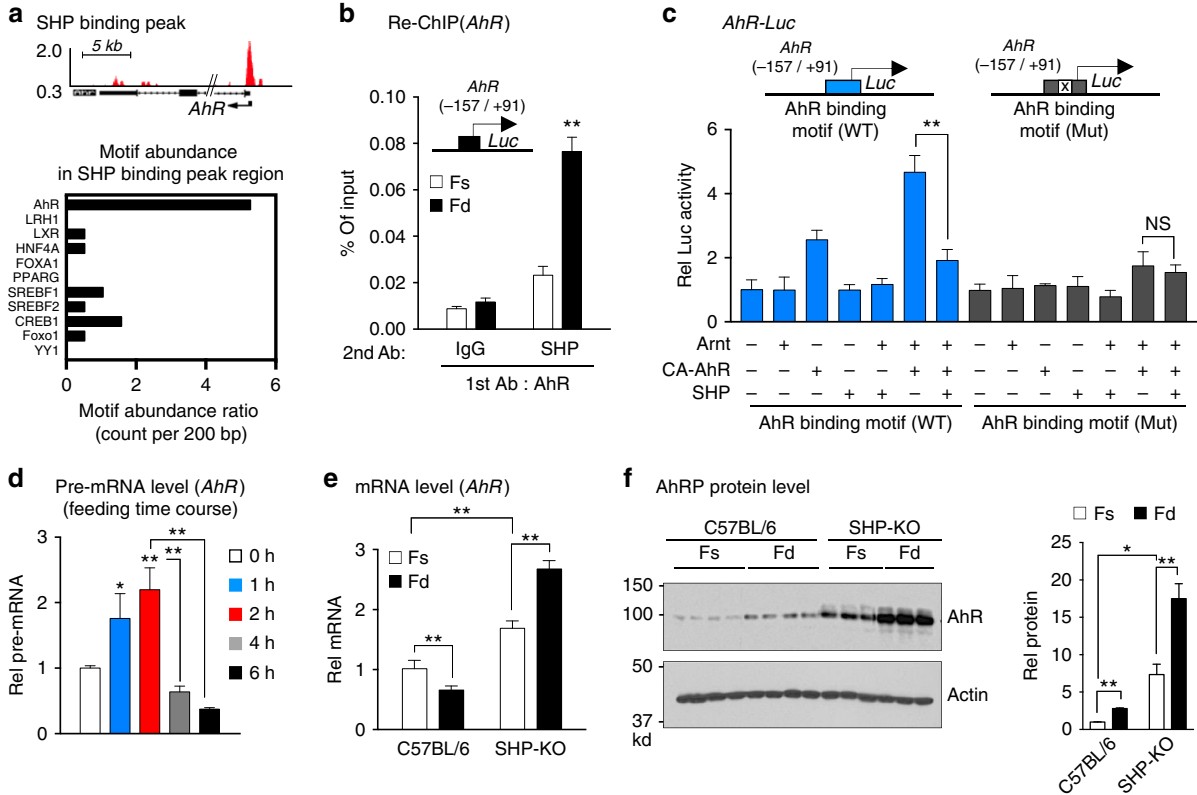

**Fig. 5** AhR induces its own expression and SHP inhibits the AhR function. **a** SHP binding peak at the AhR promoter from published ChIP data and potential binding sites for transcription factors (bottom) within SHP peak region (top) were identified using the JASPAR database. **b** Mice ($n = 5$ mice per group) were refed for 6 h after fasting overnight. Liver chromatin was immunoprecipitated with AhR antibody first and then eluted and re-precipitated with SHP antibody. **c** Hepa1c1c7 cells were transfected with indicated plasmids. After 2 days, cells were treated with FGF19 for 1 h, and luciferase activities were measured and normalized to β-galactosidase activity ($n = 5$). **d** Mice ($n = 5$ mice per group) were fasted overnight or fasted and refed for the indicated times, and AhR pre-mRNA levels were determined by qRT-PCR. **e, f** C57BL/6 control mice and SHP-KO mice ($n = 3–5$ mice per group) were fasted overnight (Fs) or fasted and refed for 6 h (Fd). Hepatic mRNA (**e**) and proteins (**f**) levels were determined. Full size immunoblots are in Supplementary Figure 12b. Means ± SD are shown, and statistical significance was measured using the **c, d** one- or **b, e, f** (right) two-way ANOVA with the FDR post-test. *$P < 0.05$, **$P < 0.01$, NS, statistically not significant

fasted mice and were further increased by feeding (Fig. 5e). Interestingly, in contrast to the effects on mRNA in control C57BL/6 mice, AhR protein levels were still increased about 2- to 3-fold at 6 h after feeding, while in SHP-KO mice, protein levels were increased by 6-fold in fasted mice compared to C57BL/6 mice and feeding increased levels 2- to 3-fold (Fig. 5f, Supplementary Figure 12b). These results indicate that AhR induces expression of its own gene early after feeding, but the expression is blunted in the late fed state in a SHP-dependent manner even though protein levels of AhR remain elevated.

**Phosphorylation of SHP is important for interaction with AhR.** Previous studies have shown that SHP is phosphorylated at Thr-55 (T55) in response to FGF15/19 signaling and this phosphorylation is important for nuclear localization of SHP and the interaction of SHP with repressive histone-modifying proteins to inhibit its target genes[19, 20]. We thus examined whether this phosphorylation was required for the inhibition of AhR-mediated transactivation of 1C genes by SHP in mice in vivo using a phosphorylation-defective T55-SHP mutant.

SHP-WT or T55A-SHP proteins were adenovirally expressed in SHP-KO mice at levels similar to those detected in C57BL/6 mice (Fig. 6a, b). In CoIP assays using whole cell extracts, interaction of SHP-WT with AhR was readily detected in FGF19-treated mice but interaction with T55A-SHP was undetectable (Fig. 6c, Supplementary Figure 12c), indicating the critical role of

FGF19-induced T55 phosphorylation of SHP in its interaction with AhR. Further, mRNA levels of AhR target genes, *Pemt*, *Gnmt*, and *AhR*, were significantly reduced in mice expressing SHP-WT compared to control mice infected with Ad-GFP, but not in mice expressing T55A-SHP (Fig. 6d). Consistent with these results, hepatic PC levels were decreased and SAM levels increased, and the ratios of PC/PE and SAH/SAM levels were decreased in mice expressing SHP-WT compared to control mice, and again these effects were blunted in mice expressing T55A-SHP (Fig. 6e). These results indicate that FGF19-induced phosphorylation of SHP at T55 is important for its interaction with AhR in regulation of hepatic PC/SAM levels in mice.

**SHP or *Pemt* downregulation reverses AhR-increased steatosis.** Maintaining a balance of PC and SAM levels is critical for hepatic lipid metabolism and imbalance of these metabolites underlies the development of NAFLD[3–6]. We observed that AhR increases expression of *Pemt*, which increases PC levels (Fig. 3d, f) and the increases are inhibited by SHP (Figs. 4–6). Further, adenoviral-mediated acute downregulation of *Pemt* decreased liver TG and neutral lipid levels in high-fat diet (HFD) mice and ob/ob mice[3] and adenoviral-mediated expression of SHP in HFD mice also resulted in decreased lipid accumulation in the liver and liver TG levels (Supplementary Figure 7). Thus, we tested whether expression of AhR contributes to the development of fatty liver in mice fed a high-fat/high fructose (HF/HF) diet and further tested

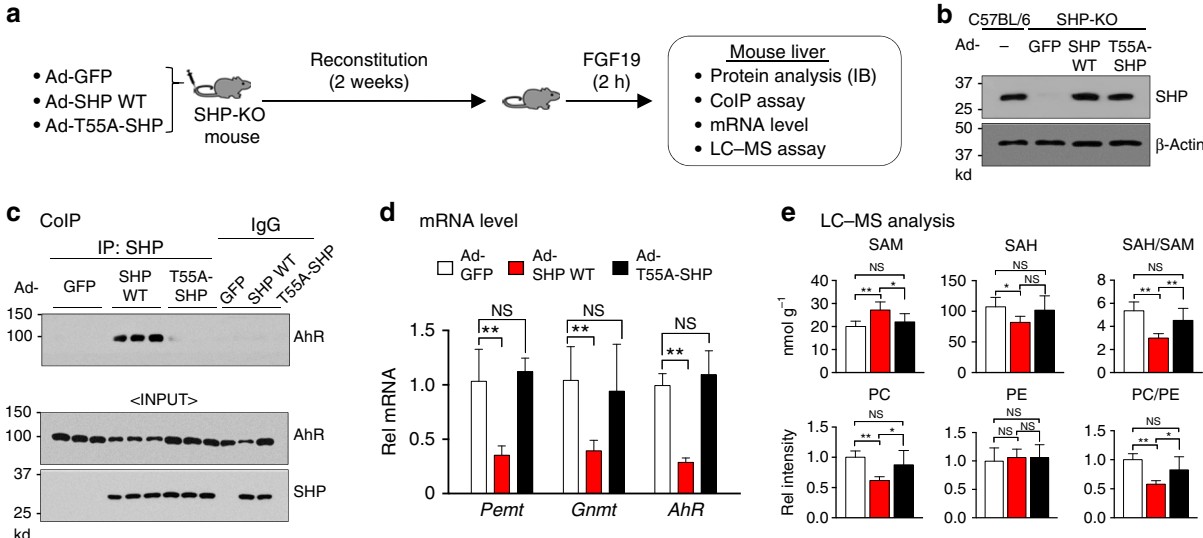

**Fig. 6** FGF19-induced phosphorylation of SHP at T55 is required for functional interaction with AhR. **a** Experimental outline: SHP-KO mice ($n = 5$ mice per group) were injected via the tail vein with control Ad-GFP, Ad-SHP-WT or p-defective Ad-T55A-SHP, and 2 weeks later, mice were treated with FGF19 for 2 h, and livers were collected. **b** Protein levels of SHP-WT or T55A-SHP in liver extracts were detected by IB. **c** The interaction between SHP-WT or T55A-SHP with AhR was determined by CoIP using whole cell liver extracts. AhR and SHP protein levels in input samples are shown. Full size immunoblots are in Supplementary Figure 12c. **d** mRNA levels of *Pemt*, *Gnmt*, and *AhR* were determined by qRT-PCR. **e** Selected 1C cycle metabolites levels were determined by LC–MS and the ratios of SAH/SAM and PC/PE were calculated. Means ± SD ($n = 5$) are shown, and statistical significance was measured using the **d**, **e** one-way ANOVA with the FDR post-test. *$P < 0.05$, **$P < 0.01$, NS, not statistically significant

whether co-expression of SHP or downregulation of *Pemt* attenuates the development.

CA-AhR and either SHP-WT or small hairpin RNA (shRNA) for *Pemt* were adenovirally expressed in obese mice that had been fed a HF/HF diet for 8 weeks (Fig. 7a). Hepatic protein levels of Pemt were increased by expression of CA-AhR, and the Pemt levels were dramatically decreased below those in control mice by co-expression of SHP or shRNA-mediated downregulation of *Pemt* (Fig. 7b, Supplementary Figure 12d). Liver size (Fig. 7c) and liver weight/body weight (Supplementary Figure 8), and neutral lipid (Fig. 7d) and TG (Fig. 7e) levels in the liver were all increased by expression of CA-AhR in the HF/HF mice. Remarkably, co-expression of SHP or downregulation of *Pemt* in these mice reversed the increases in liver size and weight and TG and neutral lipid levels below those of control mice (Fig. 7c–e) and the appearance of the liver was more normal (Fig. 7c). The ratios of PC/PE and SAH/SAM (Fig. 7g) and hepatic expression of *Pemt and Gnmt* as well as a known AhR target *Cyp1a1*[7] (Fig. 7f) were increased in mice expressing CA-AhR as expected, and these AhR-mediated increases were also reversed by co-expression of SHP or downregulation of Pemt.

These results indicate that exacerbation of liver steatosis by adenoviral-mediated expression of CA-AhR in diet-induced obese mice is dependent on the expression of *Pemt*. Further, this exacerbation and increased expression of *Pemt* can be reversed by co-expression of SHP (Fig. 7d–f), suggesting that blocking by SHP of the AhR-mediated increases in PC levels and promotion of liver steatosis contributes to the beneficial effects of SHP on fatty liver. Since acute expression of SHP alone reduces dietary-induced hepatic lipid accumulation (Supplementary Figure 7), there are likely independent beneficial effects of SHP, which is consistent with the reduction of hepatic lipid accumulation in the CA-AhR-expressing mice below that of the control mice.

**AHR and PEMT and the PC/PE ratio are altered in NAFLD.** PC plays an essential role in packaging of lipids as lipid droplets

or lipoproteins, both of which are elevated in obesity[38]. Indeed, hepatic PC levels and the ratio of PC/PE were shown to be elevated in fatty liver of obese mice exhibiting obesity-induced ER stress[3, 39]. We also observed that hepatic mRNA levels of *Pemt*, *Gnmt*, and *AhR* were elevated in diet-induced obese mice compared to lean mice fed a normal chow (Supplementary Figure 9). To determine if hepatic expression of *PEMT* and the PC/PE ratio are also aberrantly elevated in NAFLD steatosis patients, we measured mRNA levels of *PEMT*, *AHR*, and *SHP* in liver samples from 15 normal individuals, 15 simple steatosis patients, and 15 severe NASH-fibrosis patients.

Hepatic mRNA levels (Fig. 8a) and protein levels (Fig. 8b, Supplementary Figure 12e) of PEMT and AHR were significantly elevated in both mild simple steatosis and severe NASH-fibrosis patients compared to normal individuals, while SHP mRNA levels were not significantly changed. AhR in the tissue samples migrated at the same mobility as AhR from human HepG2 cells and, as expected, slightly slower than mouse AhR (Supplementary Figures 10 and 13b), which confirmed that AhR was being detected in the blots from tissue samples. Since FGF19 signaling is impaired in fatty livers of diet-induced obese mice[40] and NAFLD patients[41], we also measured protein levels of phosphorylated T55-SHP, which is induced by FGF19 signaling[19]. Intriguingly, phosphorylated T55-SHP levels were dramatically decreased over 80%, while SHP protein levels were little changed (Fig. 8b, Supplementary Figure 12e).

Consistent with increased AHR and PEMT expression in steatosis patients (Fig. 8a, b), hepatic PC levels were modestly, but significantly, increased, and PE levels were decreased resulting in an increased ratio of PC/PE (Fig. 8c). Notably, this pattern was observed for most PC and PE lipids containing either saturated or polyunsaturated fatty acids (Supplementary Figure 11). Similarly, the ratio of PC/PE was also increased in severe NASH-fibrosis patients but, intriguingly, both PC and PE levels were so dramatically decreased that the significance of the increased ratio is not clear (Fig. 8c). These results demonstrate that hepatic expression of AHR and PEMT is aberrantly elevated and the ratio

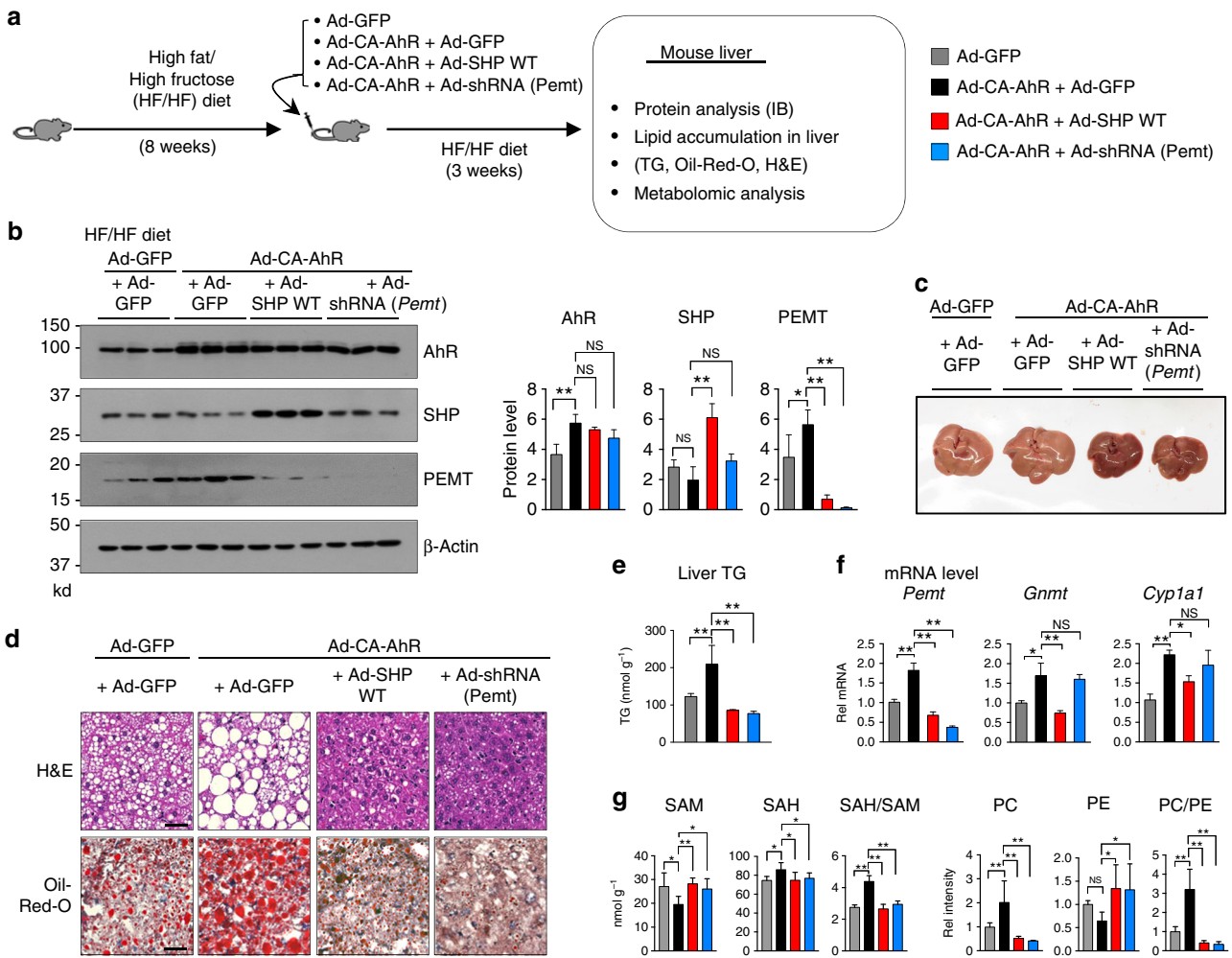

**Fig. 7** Downregulation of Pemt or overexpression of SHP blunts AhR-mediated increases in PC levels and liver steatosis in mice fed HF/HF. **a** Experimental outline: Mice ($n = 5$ mice per group) were fed a HF/HF diet for 8 weeks and infected with the indicated adenovirus. For the Ad-CA-AhR injection, Ad-GFP was added so that the same number of viral particles were injected for each group. At 3 weeks after adenoviral injection, livers were collected for further analyses. **b** Samples from three mice per each group were randomly selected and analyzed by IB (left) and quantification of the bands is shown (right). Full size immunoblots are in Supplementary Figure 12d. **c** Representative images of livers from the experimental groups. **d** Oil Red-O staining of frozen liver sections. Scale bar: 100 μM. **e** Liver TG levels. **f** mRNA levels of Pemt, Gnmt, and Cyp1a1 as determined by qRT-PCR. **g** Ratios of SAH/SAM and PC/PE as calculated from levels of these metabolites determined by LC–MS ($n = 5$). Means ± SD are shown ($n = 5$ mice), and statistical significance was measured using the **b**- (right), **e**–**g** one-way ANOVA with the FDR post-test. $*P < 0.05$, $**P < 0.01$, NS, not statistically significant

of PC/PE is significantly increased in simple steatosis and severe NASH-fibrosis patients. Notably, in severe NASH-fibrosis patients, hepatic PC and PE levels are dramatically decreased even though hepatic AHR and PEMT levels are elevated.

## Discussion

PC is a critical determinant of hepatic TG levels[3–6]. As a major component of membranes, PC plays an essential role in the packaging and secretion of TG-rich VLDL from the liver, and thus low levels of PC lead to increased liver TG levels. Low PC levels also increase liver TG levels by promoting nuclear translocation of SREBP-1, and transcriptional activation of the lipogenic genes[4]. Paradoxically however, abnormally elevated PC levels also lead to increased liver TG levels by promoting metabolism of PC into diacylglycerol (DG) and, subsequently, TG synthesis[3, 5, 6]. Thus, either abnormally low or high levels of PC induce liver steatosis, and PC levels must be tightly regulated under physiological conditions. This study identifies AhR and

SHP as novel physiological regulators that control hepatic PC levels. AhR mediates transcriptional induction of 1C genes including Pemt and increases PC levels early after feeding, and SHP blocks this AhR function in the late fed state.

It is well known that AhR mediates transcriptional responses to environmental toxicants[7, 8], but recent evidence indicates that AhR also has a role in hepatic lipid metabolism. AhR overexpression exacerbates diet-induced hepatosteatosis by upregulating CD36 and increasing fatty acid uptake into liver[14, 32], while protecting against obesity and diabetes by induction of FGF21[12], a hepatokine with lipid-lowering and insulin-sensitizing effects[28]. In contrast, AhR deficiency is beneficial for hepatosteatosis as well as obesity and diabetes[32]. Further, a role of AhR in hyperhomocysteine-induced hepatic steatosis by induction of CD36 has been reported[13]. In the present study, we find that AhR promotes liver steatosis in diet-induced obese mice through a different pathway by directly inducing Pemt, which results in an increase in the ratio of PC/PE. Our findings are consistent with previous studies showing that hepatic expression of Pemt was

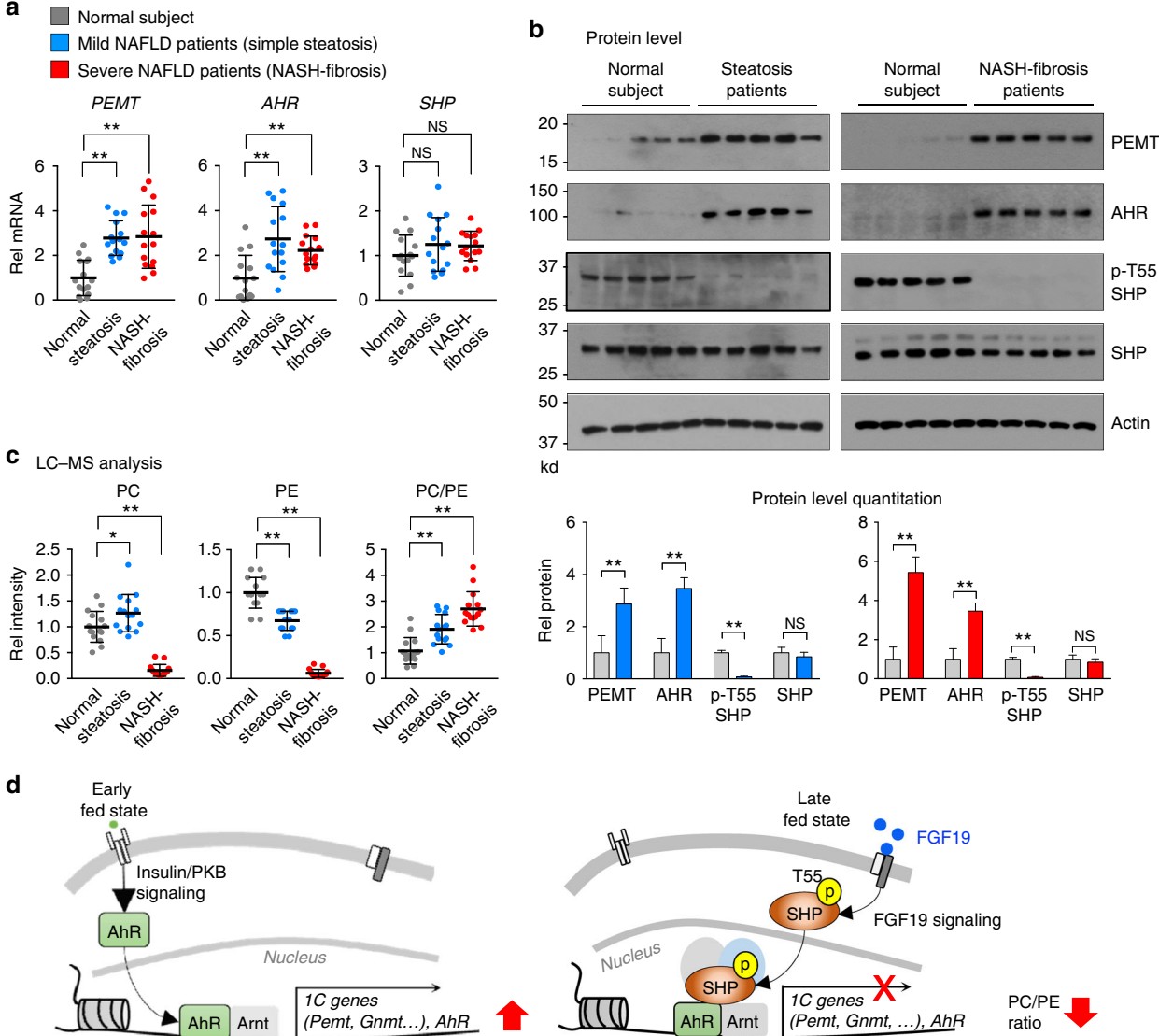

**Fig. 8** Elevated AHR, PEMT, and PC levels in livers of NAFLD steatosis and NASH patients. **a** mRNA levels of *PEMT*, *GNMT*, and *AHR* in liver samples of 15 normal, 15 simple steatosis, and 15 severe NASH-fibrosis patients were determined by qRT-PCR. **b** Protein levels of PEMT, AHR, phosphorylated T55-SHP, and SHP in liver extracts from 5 samples each randomly pooled from 3 individuals (total 15 individuals) were analyzed by IB analysis (top) and the band intensities were quantified (bottom). The same normal samples were analyzed as controls in the two separate gels. Full size immunoblots are in Supplementary Figure 12e. **c** Hepatic PC and PE levels were determined by LC–MS and the ratios of PC/PE calculated. Means ± SD are shown (*n* = 15), and statistical significance was measured using the **b** (bottom) Student's *t*-test or **a**, **c** one-way ANOVA with the FDR post-test. \**P* < 0.05, \*\**P* < 0.01, NS, not statistically significant. **d** Model: Temporal transcriptional regulation of the 1C cycle genes by a postprandial AhR–SHP axis. Nuclear levels of AhR are increased early after feeding by insulin/PKB signaling and nuclear AhR mediates transcriptional induction of 1C cycle genes, including *Pemt and AhR*, which results in increased hepatic PC and decreased SAM levels. In the late fed state, SHP blocks this AhR action through FGF15/19 signal-induced phosphorylation at T55

abnormally increased and the ratio of PC/PE was elevated in fatty livers of obese mice, whereas adenoviral-mediated acute downregulation of *Pemt* decreased liver steatosis[3]. Contrary to these results, *Pemt*-KO mice develop fatty liver[42]. This is likely due to the difference between acute and chronic downregulation of *Pemt*. In the Pemt-KO transgenic mice, secretion of VLDL from the liver is blocked in part due to chronic deficiency in PC, while in Pemt-downregulated mice, hepatic TG synthesis are decreased in part due to decreased levels of PC and DG levels[6].

The present study reveals a novel function of the postprandial FGF19-SHP pathway in the regulation of 1C cycle genes. FGF19 is a late fed-state hormone that acts during the metabolic transition from a fed to fasted state. FGF19 has received great attention as a drug target because of beneficial metabolic

outcomes after treatment with pharmacological FGF19, including lipid-lowering effects[28], but the underlying mechanisms are not clearly understood. In this study, we find that FGF19 decreases liver TG levels in part through a reduction of hepatic PC levels in a SHP-dependent manner. Mechanistically, we find FGF19-induced phosphorylation of SHP at T55 increases the interaction of SHP with AhR, resulting in the inhibition of AhR-mediated transactivation of *Pemt*. Recent studies showed that SREBP-1 increases expression of a conserved set of genes in the 1C cycle in *Caenorhabditis elegans* and mice[4] and that the PC-sensing nuclear receptor LRH-1 increases expression of *Gnmt* and regulates methyl metabolite pools in mouse liver in vivo[36]. We confirmed that *Gnmt* promoter activity was highly increased by LRH-1, while *Pemt* promoter activity was modestly but

significantly increased, and SHP inhibited the LRH-1-mediated increase (Supplementary Figure 5). Thus, although the present studies focus on the AhR–SHP interaction, SHP may also repress 1C genes by inhibiting other transcriptional activators, such as LRH-1.

The present study provides a model for temporal regulation of 1C genes by feeding-sensing AhR and SHP (Fig. 8d). Nuclear levels of AhR are increased by an insulin/PKB signaling and increase transcription of *Pemt* early after feeding, while SHP is translocated into the nucleus in response to FGF15/19 signaling, binds to AhR-bound chromatin, and blocks AhR function in the late fed state. In ChIP studies, an increase in occupancy of AhR at the *Pemt* promoter was detected as early as 1 h after feeding and peaked at 2 h, while the increase in SHP occupancy was not detected until 2 h and peaked at 4 h after feeding. Thus, it is likely that postprandial AhR and SHP regulate transcription of *Pemt* gene in a time-dependent manner. In the immediate response to feeding, possibly in response to insulin signaling, AhR is translocated into the nucleus, activates expression of *Pemt*, and increases production of PC needed for hepatic lipogenesis. In the late fed state, phosphorylation of T55 by FGF15/19 signaling[19] induces the interaction of SHP with AhR and the inhibition of AhR function, thereby reversing the increase in postprandial PC levels.

Previous studies reporting hepatic PC levels and the ratio of PC/PE in NAFLD patients are controversial, which may be due to small sample size numbers, and poor description of clinical phenotypes and disease severity[43–45]. In this study, we observed that hepatic PC levels and the PC/PE ratio are elevated in simple steatosis NAFLD patients, which is consistent with our studies of obese mice. The increased levels of AhR and PEMT likely underlies the increased PC levels. While SHP expression was not significantly altered, phosphorylation of SHP at T55 was dramatically decreased in both mild and severe NAFLD patients, suggesting FGF19 signaling is impaired in these patients as previously reported[41]. Interestingly, in severe NAFLD patients, both PC and PE levels were dramatically decreased, even though AhR and PEMT levels were elevated. PC levels, therefore, might be regulated through different pathways in mild and severe conditions. Supporting this idea, in recent global transcriptome studies, distinct gene signatures were identified in livers of severe NAFLD patients with fibrosis compared to mild steatosis patients[46], suggesting that specific metabolic and tissue repair/regeneration pathways are activated selectively in severe NAFLD patients.

In conclusion, this study identifies AhR and SHP as novel physiological regulators that control hepatic PC and SAM levels in the 1C cycle in the fed state. Maintaining the PC and SAM levels is critical for normal cellular functions and dysregulation of these metabolite levels is associated with numerous liver diseases, including fatty liver disease, cirrhosis, and hepatocellular carcinoma[2, 47]. The AhR–SHP axis may provide novel therapeutic targets for treatment of NAFLD and other diseases associated with dysregulated 1C cycle metabolism.

## Methods

**Reagents.** Antibodies for SHP (sc-30169), AhR (sc-133088), and actin (sc-1616) were purchased from Santa Cruz Biotechnology and for PEMT and ARNT from Thermo Fisher (PA5-42383) and NOVUS Biologicals (NB100-110), respectively. Texas Red goat anti-rabbit IgG was obtained from Invitrogen (T-2767). A SHP antibody specific to phospho-Thr-55 (p-T55-SHP) was produced commercially (Abmart, Inc.) and used in previous studies[19, 26] and its specificity has been described previously[19]. Endogenous AhR in hepatocytes was downregulated using the pooled small interfering RNAs for AhR (ON-TARGETplus SMART pool, GE Dharmacon, Inc.) to minimize off-target effects. Hepa1c1c7 cells were obtained from ATCC (CRL-2026).

**Genetic mouse models.** SHP-KO mice have been back-crossed for up to 10 generations to C57BL/6 mice[21, 48]. The AhR-KO mice (The Jackson Lab) are in a C57BL/6 background[12]. Age- and sex-matched C57BL/6 mice were used as

controls for SHP-KO and AhR-KO mice. For TetRE-CA-AhR and FGF15-KO mice, littermates were used as WT controls. SHP-KO mice are resistant to bile acid-induced hepatotoxicity[21, 49, 50]. CA-AhR mice have spontaneous fatty liver[14], increased sensitivity to diet-induced NASH, and HFD-induced metabolic syndrome[12, 51]. In AhR-KO mice, the immune system is impaired, hepatic fibrosis is increased, and the liver size is reduced by 50%[10].

**Animal experiments.** Male C57BL/6 mice and SHP-KO mice (8–12 weeks old) were fasted for 4–12 h and injected via the tail vein with vehicle or FGF19 (1 mg kg$^{-1}$) at 9:00 a.m., and 2 h or 6 h later, livers were collected. For feeding experiments, mice were fasted for 12 h and then fed for 6 h and livers were collected for further analyses. For FXR activation studies, mice were fasted for 12 h and then treated intraperitoneally with GW4064 (30 mg kg$^{-1}$ in corn oil) for 6 h. For dietary obese mouse studies, mice were fed normal chow or a HFD (Harlan Teklad, TD88137) with 25% of fructose (Sigma, F0127) in water for 11 weeks. Since 1C cycle metabolism is influenced by the circadian rhythm[25], mouse experiments were done at similar times of the day. AhR-KO and TetRE-CA-AHR transgenic mice[12] and SHP-KO[21] and FGF15[52] mice have been described previously. For adenoviral experiments, mice were injected via the tail vein with 0.5–1.0 × 10$^9$ active viral particles in 100 μl phosphate-buffered saline (PBS) and 2–3 weeks later, the mice were killed and livers were collected. Injection of these viral doses does not elicit marked inflammatory responses[20]. All experiments were approved by the Institutional Animal Care and Use and Biosafety Committees of the University of Illinois at Urbana-Champaign.

**LC–MS analysis.** Liver samples were minced and extracted in 0.4N perchloric acid. The 1C cycle metabolites in the liver homogenates were detected by liquid chromatography–mass spectrometry (LC–MS) analysis on a Sciex 5500 QTrap with Agilent 1200 LC (AB Sciex.) for SAM and SAH, and a Thermo Q-Exactive Hybrid Quadrupole-Orbitrap Mass Spectrometer with an Ultimate 3000 HPLC (Thermo Fisher Scientific) for PE and PC. Liver TG levels were measured using a Serum Triglyceride Determination Kit (TR0100, Sigma-Aldrich).

**Histological analysis.** Neutral lipids in frozen liver sections were detected by Oil Red-O staining, and paraffin-embedded sections were stained with hematoxylin and eosin. Stained slides were imaged with a NanoZoomer Scanner (Hamamatsu). For IF analysis, paraffin-embedded liver sections were incubated with primary antibody for 2 h (dilution 1:100) followed by the secondary antibody Texas Red goat anti-rabbit IgG (dilution 1:200 in PBS) for 1 h. Nuclei were labeled with Hoechst 33258 (dilution 1:1000, from Sigma, 861405) for 5 min. Images were taken and processed using ZEN software (Zeiss LSM700).

**DNA motif analysis.** Consensus transcription factor binding motifs within SHP binding peaks increased by FGF19 treatment were determined using the JASPAR online program (2010; http://jaspar.genereg.net/).

**Immunoblot (IB) analysis.** Liver tissues were washed with ice-cold PBS and homogenized. Cells were washed with PBS and harvested in 200 μl of RIPA buffer (50 mM Tris, pH 7.5, 1 mM EDTA, 1% NP40, 1% sodium deoxycholate, 0.1% SDS) containing protease inhibitors (1 mM phenylmethane sulfonyl fluoride, 10 μg ml$^{-1}$ leupeptin, 10 μg ml$^{-1}$ aprotinin, and 2 μM pepstatin A) and phosphatase inhibitors (1 mM NaF and 1 mM Na$_3$VO$_4$). Total cell lysates containing equal amounts of protein were subjected to electrophoresis, transferred to polyvinylidene difluoride membranes, and blocked in TBST buffer (20 mM Tris-HCl, pH 7.5, 150 mM NaCl, and 0.1% Tween-20) containing 5% non-fat milk. Blots were incubated with primary antibodies, followed by anti-rabbit or anti-mouse secondary IgG horseradish peroxidase-linked antibody.

**CoIP analysis.** Mouse liver extracts were prepared in CoIP buffer (50 mM Tris, pH 8.0, 150 mM NaCl, 2 mM EDTA, 0.3% NP40, 10% glycerol) and incubated with 2 μg of antibodies for AhR or SHP or control IgG as indicated for 30 min, and 35 μl of 25% protein G agarose was added. After 2 h, agarose beads were washed with the CoIP buffer and bound proteins were detected by IB.

**ChIP and re-ChIP analyses.** For ChIP, minced liver tissues or cells were washed twice with PBS, then incubated with 1% (w/v) formaldehyde for 10 min at 37 °C. Glycine was added to 125 mM for 5 min at room temperature. Chromatin solutions in sonication buffer (50 mM Tris-HCl, pH 8.0, 2 mM EDTA, and 1% SDS) with protease inhibitors were sonicated four times with 10 s intervals using a QSonica XL-2000 instrument at power output setting 8 or using a QSonica 800R2-110 at amplitude setting 70% with sonication pulse rate 15 s on and 45 s off. Then, chromatin was immunoprecipitated with 2 μg antibody for SHP, AhR, ARNT, or control IgG overnight at 4 °C with rotation. The immune complexes were collected by incubation with a Protein G–Sepharose slurry (Invitrogen) containing salmon-sperm DNA for 1 h, washing with 0.1% SDS, 1% Triton X-100, 2 mM EDTA, 20 mM Tris-HCl, pH 8.0, three times containing successively 150 mM NaCl, 500 mM NaCl, or 0.25 M LiCl, and then incubated overnight at 65 °C to reverse the crosslinking. For re-ChIP, chromatin was immunoprecipitated with AhR antibody

first by binding to Protein G–Sepharose and washing as described for ChIP. Immunoprecipitated chromatin was eluted by incubation with 10 mM dithiothreitol for 30 min at 37 °C, the sample was diluted 20× by addition of dilution buffer (20 mM Tris-HCl, pH 8.0, 2 mM EDTA, 150 mM NaCl, and 1% Triton X-100), and chromatin fragments were re-precipitated using antibodies for ARNT or SHP. Then, crosslinks were reversed, and DNA was isolated for qPCR. Primer sets used are shown in Supplementary Table 1.

**Construction of Ad-CA-AhR**. Human CA-AhR complementary DNA from pCMX-CA-AhR[12] was cloned into the Ad-Track cytomegalovirus (CMV) vector, and Ad-CA-AhR was produced using the AdEasy[TM] System. Amino acids 273 to 432 of human AhR are deleted in CA-AhR so that the molecular weight of human CA-AHR is similar to that of mouse WT AhR. For the construction of Ad-shRNA for *Pemt*, a shRNA for *Pemt* with the U6 promoter (Cyagen Biosciences Inc.) was cloned into the Ad-Track vector.

**Luciferase reporter assay**. Genomic DNA fragments at the promoter of *Pemt* (−190/+156), *Gnmt* (−197/+278), and *AhR* (−157/+91) that contain SHP peaks[26] were amplified by PCR from mouse genomic DNA and were inserted into the pGL3-basic-luc plasmid. The AhR binding motif was mutated using the Site-Directed Mutagenesis Kit (#200521, Agilent Technologies). Hepa1c1c7 were cotransfected with 200 ng of reporter plasmids containing the *Pemt, Gnmt*, or *AhR* promoter regions, 100 ng of CMV β-galactosidase plasmid, and 50 ng Arnt, 5 ng AhR or CA-AhR, and 5 to 50 ng SHP expression plasmids. Luciferase activities were normalized to β-galactosidase activities.

**qRT-PCR analysis**. RNA was isolated from liver or hepatocytes and quantified by quantitative reverse transcription-PCR (qRT-PCR), normalized to 36B4 mRNA. Primer sequences for qRT-PCR are shown in Supplementary Table 2.

**Subcellular localization studies**. Hepa1c1c7 cells were cultured in 5 mM D-glucose (low-glucose) and serum-free media overnight and then treated with insulin (100 nM), FGF19 (50 ng/ml), CDCA (50 μM) for 15 min, or inhibitors of either AKT/PKB (PKB124005, 5 μM) or ERK (PD98059, 20 μM) signaling for 30 min prior to insulin treatment and then cells were harvested. Cells were resuspended in hypotonic buffer (10 mM Hepes, 1.5 mM MgCl2, 10 mM KCl, 0.2% NP40, 1 mM EDTA, and 5% sucrose) and lysed by homogenization. Nuclei were pelleted by cushion buffer (10 mM Tris-HCl, pH 7.5, 15 mM NaCl, 60 mM KCl, 1 mM EDTA, and 10% sucrose). The nuclear pellet and cytoplasmic supernatant were collected after centrifugation with 5000 rpm for 3 min, and fractionation quality was monitored by detection of lamin and tubulin levels by IB.

**Analyses of NAFLD patient samples**. Frozen unidentifiable liver samples of 15 normal individuals, 15 simple steatosis patients, and 15 NASH-fibrosis patients were obtained from the Liver Tissue Procurement and Distribution System that operates under a contract from the National Institutes of Health and ethical approval was not required. Hepatic mRNA, protein, and selected 1C metabolite levels were determined by qRT-PCR, IB, and LC–MS, respectively.

**Statistical analysis**. GraphPad Prism 6 (GraphPad software version 6.01) was used for data analysis. Statistical significance was determined by Student's two-tailed *t*-test or one- or two-way analysis of variance (ANOVA) with the false discovery rate (FDR) post-test for single or multiple comparisons as appropriate. *P*-values < 0.05 were considered statistically significant.

**Data availability**. The data that support the findings of this study are available from the corresponding author upon reasonable request.

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

## Acknowledgements

We thank Eric H. Xu for providing recombinant FGF19 and David Moore, Li Wang, and Sayee Anakk for providing SHP-KO mice. We thank the Liver Tissue Procurement and Distribution System of the NIH for providing human liver specimens of NAFLD patients. We also thank Lucas Li, Director of Metabolomics Center at UIUC, for assistance with the LC–MS analysis. This study was supported by an American Heart Association Scientist Development Award (16SDG27570006) to Y.-C.K., an American Heart Association post-doctoral fellowship (17POST33410223) to S.B., and by grants from the National Institutes of Health (DK62777 and DK95842) and the American Diabetes Association (1-16-IBS-156) to J.K.K.

## Author contributions

Y.-C.K and J.K.K. designed research; Y.-C.K., S.S., and S.B. performed experiments; Y.-C.K., S.S., S.B., B.Ko., B.Ke., and J.K.K. analyzed data; Y.Z. and J.M. performed bioinformatics analysis; G.G. and W.X. provided key materials for the study; and Y.-C.K., B.Ke., and J.K.K. wrote the paper.

## Additional information

**Competing interests:** The authors declare no competing financial interests.

