## [Peer Review File · Nature Communications]

Editorial Note: Parts of this peer review file have been redacted as indicated to remove third-party material where no permission to publish could be obtained.

Reviewer #1 (Remarks to the Author):

This is a paper from Kim et al that examined the roles of aryl hydrocarbon receptor (AhR) and SHP in the regulation of 1C metabolism genes expression and the levels of S-adenosylmethionine (SAM) and phosphatidylcholine (PC) in response to feeding. Using a variety of in vitro and in vivo models, the key findings are that 1) AhR and SHP both bind to the promoter region of multiple genes involved in 1C metabolism (i.e. *Pemt*, *Gnmt*, *Ahcyl2*, *Mthfr*, *Mtfmt*), 2) early after feeding, AhR nuclear content increases in response to insulin-PKB signaling and this induces the expression of these genes, likely at the promoter level, 3) in contrast, SHP is induced later after feeding by FGF15/19 signaling, leading to phosphorylation of SHP at Thr-55, increased nuclear content and interaction with AhR to shut down the same genes. At the metabolite level, increased AhR activity resulted in lower SAM, higher SAH, higher PC and lower PE levels and increased SHP activity resulted in the exact opposite. Overexpressing constitutively active AhR worsened fat accumulation after 8 weeks of HF/HF diet, and this was prevented if SHP was overexpressed or *Pemt* was silenced concurrently. In 15 each of normal, NAFLD and NASH patients, authors found higher *PEMT*, *AHR* expression but lower p-T55-SHP in both NAFLD and NASH patients. PC/PE was higher in both NAFLD and NASH patients, although both PC and PE absolute levels were markedly reduced in NASH patients. Authors concluded that they have identified AhR and SHP as novel physiological regulators of hepatic PC and SAME levels during feeding and suggested the AhR-SHP axis may provide novel therapeutic targets for the treatment of NAFLD.

Major comments:

1. For all experiments comparing WT to KO or TG, describe whether the WT mice are littermates.
2. Authors should briefly describe the phenotypes of SHP KO, CA-AhR transgenic, and AhR-KO mice.
3. Figure 3C – The same experiment should be done to verify *Gnmt*, *Ahcyl2*, *Mthfr* and *Mtfmt* are AhR targets.
4. Figure 4a and b – experiment was done using a mouse hepatoma cell line. This should be repeated using primary human hepatocytes to ensure the same response occurs.
5. Do SHP and AHR directly interact? Co-IP does not address this. This should be evaluated.
6. Figure 5C – what is the effect of adding SHP to the Luc activity of AhR WT and mutant motif (right graph)?
7. Figure 6 – lack of interaction between T55A-SHP and AHR may be simply due to T55A-SHP's inability to enter the nucleus. Nuclear level of SHP should be shown. The current data does not address whether SHP and AHR interaction requires phosphorylation at T55 of SHP since only nuclear proteins were evaluated.
8. Page 13 – authors state "AhR deficiency protected against HFD-induced obesity". This is the exact opposite of what was shown in reference 14, which showed AHR TG mice were protected against HFD-induced obesity and insulin resistance despite having severe fatty liver. This was attributed to enhanced FGF21, which is also an AHR target gene.
9. Figure 7 – In order to properly compare the different effects of these proteins, inclusion of at least Ad-SHP alone and Ad-shRNA (*Pemt*) alone groups (without AhR) is needed. Otherwise, all that can be concluded is that the constitutively active AhR-driven hepatic steatosis can be ameliorated by overexpressing SHP or knocking down *PEMT*. It is more important to know the roles of SHP and *PEMT* in fatty liver that develops during HF/HF diet. This also emphasizes the need of the authors to reconcile current findings against a body of literature from the *Pemt* knockout mice, which develop fatty liver.
10. Figure 8 - As the authors acknowledged, some of the literature on PC, PE levels and PC/PE in human NAFLD and NASH conflict with findings in this paper. PC and PE are made up of multiple different molecular species. Since authors used LC/MS, individual species should be shown to better understand the differences. Particularly, the authors should focus in PUFA PE and PC lipids, like those containing FA(22:6).
11. It may be that in the setting of high PC/PE, which is known to trigger ER stress and contribute to fatty liver development, lowering of *PEMT* expression is protective (as shown in reference 4). However, that does not automatically translates to targeting *PEMT* in the treatment of NASH. The

exact opposite finding – namely reduced PEMT mRNA levels were reported in NASH patients (Scientific Reports 6, article number:21721, 2016).

Minor comments:

1. Page 4 – GNMT does not catalyze the conversion of PE to PC.
2. Page 4 – The sentence “it is largely unknown how PC and SAM levels are regulated under physiological conditions and how these metabolites are dysregulated in NAFLD” is misleading as there is a large body of literature on how these metabolites are regulated and dysregulated in NAFLD.
3. Figure 1 – authors should comment on whether FGF19 (1mg/kg) is a physiological dose. Fig. 1e – are the levels significantly different between WT and KO?
4. Define Fs and Fd in legends.

Reviewer #2 (Remarks to the Author):

In this manuscript, Kim et al. investigated the regulatory cross-talk between AhR, FGF19 and SHP during the fed state and its impact on hepatic SAM/PC levels. In a previous Chip-seq-based study, the authors already showed that SHP acts as a transcriptional partner of transcription factors involved in important metabolic pathways such as cholesterol biosynthesis. In the present manuscript, the authors use the existing Chip-seq data to demonstrate that FGF19-activated SHP acts as a co-repressor of the nuclear receptor AhR. The authors demonstrate that this new Shp-AhR axis is a postprandial physiological regulator of PC/SAM levels.

Overall, the study is well designed and conducted. Its main novelty resides on the identification of insulin-activated AhR as a key regulator of genes involved in the one-carbon cycle (specifically Pemt) in the early-fed liver. Transcription of these genes is later repressed by a new FGF19-SHP-AhR axis to limit PC and SAH production. Since the balance of metabolites derived from the one-carbon cycle is crucial for hepatic function in general and for liver TG content in particular, the repercussions of these findings are extended to a NAFLD model.

Major comments:

1. The authors show significant Chip-seq peaks when SHP antibody (fig 1 and 5) is used in the promoter region of key 1C-cycle genes. This experiment lacks a proper control, like IgG, to demonstrate the specificity of the peaks. In fact, analysis of the referred Chip-Seq dataset show that the peaks presented in the manuscript are also observed in the input alone. This concern should be addressed. The same remarks hold true for the Chip-qPCR assays where the authors should show the results obtained with IgG alone or with a negative control like Actin or GAPDH.
2. Commercially available antibodies raised against SHP are notoriously known to be “difficult”. This concern should systematically be taken into account when designing experiments using such reagents. In order to confirm the specificity of their IPs in the various SHP ChIP-qPCR experiments presented in this manuscript, the authors should use chromatin isolated from the livers of Shp KO animals. Any enrichment detected in WT animals should be lost with Shp-KO mice and would satisfy the legitimate concern of any potential reader from the Shp field. The same precaution should be taken with the SHP ChIP assays using FGF19 treatment.
3. The authors interchangeably use WT and C57BL6, and it is not clear whether the WT are littermates of the KO animals, or just C57BL6 mice. It is not preferable to use C57BL6 mice as controls for the various KO mice, because of various reasons, including differences in background, microbiota, etc.. The gold standard is still to use control littermates for every mutant line.
4. Did the authors exclude the possibility that increased NR5A2 transcriptional activity may account

for the increase in 1C cycle transcripts as was previously published (PMID 26267291). Although here the authors demonstrate low NR5A2 binding motifs within the SHP peaks, it could be sufficient to affect the expression of the 1C cycle genes, especially in the absence of its transcriptional repressor SHP. In addition, it would be informative if the sequence of binding motifs of all TFs in Fig 3A are listed in the manuscript.

5. In Fig. 7A mice were either injected with an adenovirus encoding GFP alone or with a double adenovirus injection (Ad-CA-Ahr + Ad-SHP WT). Does this mean that these mice received double the number of viral particles?

6. The authors show in the last figure that PC and PE levels are dramatically downregulated, but PC/PE ratio increased in severe NAFLD (NASH) patients. The relative levels of PC and PE being so low, how would an increase in this ratio be relevant?

Minor comments:

1. In the graphs, Fs and Fd are used without indication of what it means.

Reviewer #3 (Remarks to the Author):

This manuscript is well written and is on a timely and important topic. The data is nicely presented and mostly supports the stated conclusions that significantly advances this field of research. However, there are a number of issues that need to be addressed to have the data firmly support the conclusions.

1. In figure 4b the authors have used PD98059 as an inhibitor of MAP kinase pathway and there appears to be a partial decrease in nuclear AhR levels, this can be explained by the fact that PD98059 is an AhR antagonist (Reiners, JJ, et al. 1998. Mol Pharmacol, 53, 438-45), this information needs to be added to the manuscript.

2. In figure 4b the authors have shown that the AhR is retained in the nucleus after insulin addition in Hepa 1c1c7 cells. However, whether the AHR is heterodimerized with ARNT has not been established. Since the ability of insulin to activate the AhR is a key point (e.g. Fig. 8d) then the author need to illustrate that AHR/ARNT heterodimer is actually formed by performing a gel shift analysis. Also does insulin cause Cyp1a1 activation in the Hepa 1c1c7 cells should also be included.

3. Comments about experiments in figure 8:

A. The human AhR has a molecular weight of 105 kDa that is significantly higher than the mouse AhR (i.e. 95 kDa). Yet the western blots of the CA-AhR and the human AhR show the same molecular weight, the authors need to examine the molecular weight standards and how they were placed on the western blots. In addition, in the human AHR liver sample western blot data the authors need a positive control from a human cell line to ensure that the band they are detecting actually is the AhR.

B. The human AhR is very susceptible to proteolysis yet the western blot for the human liver samples does not show any degradation products. In the methods section, how the liver samples were prepared needs to be added to the methods, were protease inhibitors added.

C. The mRNA data for the Ahr in the 15 human liver samples shows a wide range of variation in

the steatosis samples, yet the 5 samples shown in the western blot were quite consistent. Did the authors select a specific subset of samples to analyze by western blot analysis?

4. More information about the various mouse models should be added to the methods section, are they all congenic on a C57BL6/J background?

5. More details of the actual ChIP protocol should be included so that someone could actually repeat the experient.

Response to Reviewers

Reviewer #1 (Remarks to the Author):

This is a paper from Kim et al that examined the roles of aryl hydrocarbon receptor (AhR) and SHP in the regulation of 1C metabolism genes expression and the levels of S-adenosylmethionine (SAM) and phosphatidylcholine (PC) in response to feeding. Using a variety of in vitro and in vivo models, the key findings are that 1) AhR and SHP both bind to the promoter region of multiple genes involved in 1C metabolism (i.e. Pemt, Gnmt, Ahcy12, Mthfr, Mtfmt), 2) early after feeding, AhR nuclear content increases in response to insulin-PKB signaling and this induces the expression of these genes, likely at the promoter level, 3) in contrast, SHP is induced later after feeding by FGF15/19 signaling, leading to phosphorylation of SHP at Thr-55, increased nuclear content and interaction with AhR to shut down the same genes. At the metabolite level, increased AhR activity resulted in lower SAM, higher SAH, higher PC and lower PE levels and increased SHP activity resulted in the exact opposite. Overexpressing constitutively active AhR worsened fat accumulation after 8 weeks of HF/HF diet, and this was prevented if SHP was overexpressed or Pemt was silenced concurrently. In 15 each of normal, NAFLD and NASH patients, authors found higher PEMT, AHR expression but lower p-T55-SHP in both NAFLD and NASH patients. PC/PE was higher in both NAFLD and NASH patients, although both PC and PE absolute levels were markedly reduced in NASH patients. Authors concluded that they have identified AhR and SHP as novel physiological regulators of hepatic PC and SAME levels during feeding and suggested the AhR-SHP axis may provide novel therapeutic targets for the treatment of NAFLD.

Major comments:

1. For all experiments comparing WT to KO or TG, describe whether the WT mice are littermates.

Response: We have added information in supplemental methods on the WT controls used for each transgenic mouse line. The SHP-KO mice have been back-crossed for up to 10 generations to C57BL/6 mice to create mice congenic to C57BL/6 mice (Li Wang, Dev Cell, 2002; Anakk et al. J Clin Invest. 2011 121: 86–95). The AhR-KO mice (liver samples supplied by Dr. Wen Xie) are also in a C57BL/6 background (Lu P et al., Hepatology, 2015), and age-matched C57BL/6 male mice were used as controls for the SHP-KO and AhR-KO mice. For the samples from TetRE-CA-AhR mice received from Dr. Xie (Lee et al., Gastroenterology, 2010), littermates were used as controls. Likewise, for the samples from FGF-15 KO mice, received from Dr. Grace Guo (Kong et al., Hepatology, 2012), littermates were used as controls. The experiment for AhR-KO mice shown in Fig. 3e and 3g was repeated using C57BL/6 mice as WT controls and the figure was revised with the new results, which were consistent with the original results.

2. Authors should briefly describe the phenotypes of SHP-KO, CA-AhR transgenic, and AhR-KO mice.

Response: The phenotypes for the transgenic mice have been described in previous papers and we have added brief descriptions to the supplemental information in the revised manuscript as follows, "SHP-KO mice are resistant to bile acid-induced hepatotoxicity (Wang et al., Dev Cell, 2002; Kerr et al., Dev Cell, 2002, Wang et al., JBC, 2003). CA-AhR mice have spontaneous fatty liver (Lee et al., Gastroenterology in 2010), increased sensitivity to methionine-choline deficient diet-induced NASH (He et al., MCB in 2013), and protection from high-fat diet-induced metabolic syndrome (Lu et al., Hepatology, 2015). In AhR-KO mice, the

immune system is impaired, hepatic fibrosis is increased, and the liver size is reduced by 50 percent. (Fernandez-Salguero, P et al, Science in 1995).”

3. Figure 3C – The same experiment should be done to verify *Gnmt*, *Ahcy12*, *Mthfr* and *Mtfmt* are AhR targets.

Response: We agree that it is important to show that more than one 1-C gene is directly an AhR target and have added experiments with the *Gnmt* promoter to Fig. 3C and Fig. S5. *Gnmt* catalyzes the transfer of a methyl group from SAM to glycine to form SAH and sarcosine and plays a major role in regulating the ratio SAM and SAH and, thus, PC/PE levels (Maite Martinez-Una et al., Hepatology, 2013), so that we focused on *Gnmt* as a second gene. Other enzymes, noted by the reviewer, are also important, but we feel the observations that AhR occupancy at the genes of these other proteins is increased by feeding and that mRNA levels of the genes are increased and decreased in CA-AhR-TG mice and AhR-KO mice, respectively, provide good evidence that they are AhR targets.

Fig. 3. AhR transactivates 1C cycle genes and increases hepatic PC levels. (c) Hepa1c1c7 cells were transfected with a *Pemt-luc* or *Gnmt-luc* construct containing the WT or mutated AhR binding site in the *Pemt* promoter along with expression plasmids as indicated, treated with FGF19 for 2 h and luciferase activity was measured. Means +/- SD are shown (n=5), and statistical significance was measured using the one-way ANOVA with the FDR post-test. *P<0.05, **P<0.01, and NS, not statistically significant.

4. Figure 4a and b – experiment was done using a mouse hepatoma cell line. This should be repeated using primary human hepatocytes to ensure the same response occurs.

Response: We agree with the reviewer’s comment that insulin regulation of AhR should be confirmed in a human cell. While primary hepatocytes would be the best human model, we have repeated these experiments in the more convenient human HepG2 cell line, which is now shown in Fig. S3. Further, we analyzed nuclear localization of ARNT, the DNA binding partner of AhR, in these studies. Similar results were obtained with the mouse and human cells.

a Subcellular localization (HepG2)

b Subcellular localization (HepG2)

Figure S3. (a) Human HepG2 cells were grown in low-glucose and serum-free media for 12 h, and changed to complete medium, or treated with insulin, FGF19, or CDCA for 15 min, then cells were harvested. (b) HepG2 cells were pre-treated with inhibitors as indicated for 30 min prior to insulin treatment for 15 min. Protein levels of AHR and SHP in the cytoplasmic and nuclear fractions were determined by IB.

5. Do SHP and AHR directly interact? Co-IP does not address this. This should be evaluated.

Response: A previous study has shown that SHP directly interacts with ARNT using GST-pull down assays (Klinge CM et al., Arch Biochem Biophys, 2001). We, therefore, have not examined the direct interaction of SHP with AhR since SHP would be associated with the AhR/ARNT complex through ARNT.

6. Figure 5C – what is the effect of adding SHP to the Luc activity of AhR WT and mutant motif?

Response: We have redone these experiments including samples with expression of SHP alone and SHP with Arnt and replaced the original figure 5C. SHP did not affect luciferase activity alone or together with Arnt, but as shown originally, inhibited transactivation by AhR. There was little change in luciferase activity with the mutated AhR motif.

c AhR-Luc

Fig. 5. AhR induces its own expression and SHP inhibits the AhR function. (c) Hepa1c1c7 cells were transfected with indicated plasmids. After 2 days, cells were treated with FGF19 for 1 h, and luciferase activities were measured and normalized to β -galactosidase activity. Means \pm SD (n=5) are shown, and statistical significance was measured using one-way ANOVA with the FDR post-test. *P<0.05, **P<0.01, and NS, statistically not significant.

7. Figure 6 – lack of interaction between T55A-SHP and AHR may be simply due to T55A-SHP's inability to enter the nucleus. Nuclear level of SHP should be shown. The current data

does not address whether SHP and AHR interaction requires phosphorylation at T55 of SHP since only nuclear proteins were evaluated.

Response: We apologize for the oversight. Whole cell extracts were used for the Co-IP studies, which has been corrected in the manuscript. Since AhR is partially cytoplasmic, the lack of co-precipitation of T55A-SHP with AhR suggests that the mutation prevents SHP from interacting with the AhR complex. Previously, we showed that SHP phosphorylation at T55 is required for its nuclear translocation and interaction with LSD1 histone demethylase (Seok et al, JBC, 2013; Kim et al, Nature Communications, 2015) which, as suggested by the reviewer, is likely the primary effect of the mutation in blocking transcriptional regulation by SHP in the nucleus.

8. Page 13 – authors state “*AhR deficiency protected against HFD-induced obesity*”. This is the exact opposite of what was shown in reference 14, which showed AHR TG mice were protected against HFD-induced obesity and insulin resistance despite having severe fatty liver. This was attributed to enhanced FGF21, which is also an AHR target gene.

Response: As the reviewer notes, the effects of AhR on liver steatosis, obesity, and diabetes are complex. AhR overexpression exacerbates diet-induced hepatosteatosis while seemingly paradoxically protects against obesity and diabetes by induction of FGF21 (Lu, P. et al., Hepatology, 2015). AhR deficiency is beneficial for hepatosteatosis as well as obesity and diabetes (Xu, C.X. et al., Int J Obes, 2015). Therefore, while opposite effects on hepatosteatosis are observed, the similar beneficial effects of overexpression and AhR deficiency on obesity and diabetes result from different mechanisms. Our summary of the results may have been oversimplified so that this sentence has been reworded. The sentence “Previous studies have shown that AhR promotes liver steatosis in high fat diet (HFD) mice, and AhR deficiency protected against HFD-induced obesity^{14, 15}.” **has been reworded to** “AhR overexpression exacerbates diet-induced hepatosteatosis by upregulating CD36 and increasing fatty acid uptake into liver^{14, 32}, while protecting against obesity and diabetes by induction of FGF21¹², a hepatokine with lipid-lowering and insulin-sensitizing effects²⁸. In contrast, AhR deficiency is beneficial for hepatosteatosis as well as obesity and diabetes³².”

9. Figure 7 – In order to properly compare the different effects of these proteins, inclusion of at least Ad-SHP alone and Ad-shRNA (Pemt) alone groups (without AhR) is needed. Otherwise, all that can be concluded is that the constitutively active AhR-driven hepatic steatosis can be ameliorated by overexpressing SHP or knocking down PEMT. It is more important to know the roles of SHP and PEMT in fatty liver that develops during HF/HF diet. This also emphasizes the need of the authors to reconcile current findings against a body of literature from the Pemt knockout mice, which develop fatty liver.

Response: We have focused on the adenoviral-mediated acute effects of SHP and shRNA for PEMT in the context of this paper on AhR-driven steatosis, but agree with the reviewer that the SHP effects on liver steatosis is important. We have, therefore, examined the adenoviral-mediated expression of Ad-SHP WT alone on high fat diet-induced lipid accumulation in the liver (new data in Fig. S7). Expression of SHP reduces the accumulation of lipid in dietary obese mice, as well as, in mice expressing CA-AhR.

Figure S7. Mice (n=5/group) were fed normal or high-fat diet for 14 weeks and Ad-GFP control or Ad-WT of SHP were tail vein injected and maintained for 2 weeks more (total 16 weeks), and livers were isolated. (a) Effects of SHP overexpression on lipid regulation in liver were determined by liver histology detected by Oil-Red-O staining and hematoxylin and eosin (H&E) staining. (b) Triglyceride (TG) level in mouse liver was measured. Means +/- SD (n=5 mice) are shown, and statistical significance was measured using by the Student's t-test. *P<0.05, **P<0.01, and NS, statistically not significant.

In case of Ad-shRNA for *Pemt* alone, Hotamisligil and his colleagues have previously reported that lipid accumulation in liver is decreased in both *ob/ob* mice and in HFD-induced obese mice by adenoviral-shRNA for *Pemt*, which resulted in reducing ER stress (Figure from Fu et al., Nature, 2011, below). We have discussed in the Discussion the difference between these results, which are consistent with our findings with AhR, in adenoviral-mediated acute downregulation of *Pemt* and the results with chronic downregulation in *Pemt*-KO mice, which have the opposite effects on fatty liver.

10. Figure 8 - As the authors acknowledged, some of the literature on PC, PE levels and PC/PE in human NAFLD and NASH conflict with findings in this paper. PC and PE are made up of multiple different molecular species. Since authors used LC/MS, individual species should be shown to better understand the differences. Particularly, the authors should focus in PUFA PE and PC lipids, like those containing FA (22:6).

Response: We now provide analysis of all PC/PE species identified by LC/MS (Fig. S11) and have modified the discussion of the results noting the effects on PUFA lipids.

Figure S11 Hepatic individual PC and PE levels in liver samples of 15 normal, 15 simple steatosis, and 15 severe NASH-fibrosis patients were determined by LC-MS. Means +/- SD (n=15 human) are shown.

11. It may be that in the setting of high PC/PE, which is known to trigger ER stress and contribute to fatty liver development, lowering of PEMT expression is protective (as shown in reference 4). However, that does not automatically translate to targeting PEMT in the treatment of NASH. The exact opposite finding – namely reduced PEMT mRNA levels were reported in NASH patients (Scientific Reports, article number:21721, 2016).

Response: The data in Nakatsuka et al (Scientific Reports, 2016) are not directly comparable to our data since they did not include normal subject controls. Their conclusion was that “PEMT mRNA expression in liver of NASH patients was significantly lower than in NAFLD patients”. The decrease was about 20%. While no change in PEMT mRNA levels between steatosis (NAFLD) and NASH patients was observed in our samples this could be due to experimental variation or differences in the diets of US patients vs. the Japanese patients. The increase in PEMT mRNA levels in NASH patients compared to normal controls is about 3-fold, so even if our value for the NASH patients is 20% high, it would not change the conclusion that mRNA levels are increased in NASH patients.

Minor comments:

1. Page 4 – GNMT does not catalyze the conversion of PE to PC.

Response: We have corrected this mistake and revised Fig. 1a to make this clear.

2. Page 4 – The sentence “it is largely unknown how PC and SAM levels are regulated under physiological conditions and how these metabolites are dysregulated in NAFLD” is misleading as there is a large body of literature on how these metabolites are regulated and dysregulated in NAFLD.

Response: We agree and have altered this sentence.

3. Figure 1 – authors should comment on whether FGF19 (1mg/kg) is a physiological dose. Fig. 1e – are the levels significantly different between WT and KO?

Response: We have noted that the dose of FGF19 used in the experiments in Fig. 1 is a pharmacological dose. The FGF-15 KO mice studies in Fig. 2 provide evidence for physiological regulation by FGF15/19. In response to the reviewer's comment, we have now indicated whether differences between the FGF19-treated WT and KO mice are significant in Fig. 1f (1e in the original manuscript) as well as in Fig. 2b, and 2d.

4. Define Fs and Fd in legends.

Response: We have defined Fs and Fd in the legends.

Reviewer #2 (Remarks to the Author):

In this manuscript, Kim et al. investigated the regulatory cross-talk between AhR, FGF19 and SHP during the fed state and its impact on hepatic SAM/PC levels. In a previous Chip-seq-based study, the authors already showed that SHP acts as a transcriptional partner of transcription factors involved in important metabolic pathways such as cholesterol biosynthesis. In the present manuscript, the authors use the existing Chip-seq data to demonstrate that FGF19-activated SHP acts as a co-repressor of the nuclear receptor AhR. The authors demonstrate that this new Shp-AhR axis is a postprandial physiological regulator of PC/SAM levels.

Overall, the study is well designed and conducted. Its main novelty resides on the identification of insulin-activated AhR as a key regulator of genes involved in the one-carbon cycle (specifically *Pemt*) in the early-fed liver. Transcription of these genes is later repressed by a new FGF19-SHP-AhR axis to limit PC and SAH production. Since the balance of metabolites derived from the one-carbon cycle is crucial for hepatic function in general and for liver TG content in particular, the repercussions of these findings are extended to a NAFLD model.

Major comments:

1. The authors show significant Chip-seq peaks when SHP antibody (fig 1 and 5) is used in the promoter region of key 1C-cycle genes. This experiment lacks a proper control, like IgG, to demonstrate the specificity of the peaks. In fact, analysis of the referred Chip-Seq dataset show that the peaks presented in the manuscript are also observed in the input alone. This concern should be addressed. The same remarks hold true for the Chip-qPCR assays where the authors should show the results obtained with IgG alone or with a negative control like Actin or GAPDH.

Response: We had done the IgG controls for the initial ChIP assays, but did not include them in the original figures to save space. They are now added in Fig. 1c, 3b, and 4h.

2. Commercially available antibodies raised against SHP are notoriously known to be "difficult". This concern should systematically be taken into account when designing experiments using such reagents. In order to confirm the specificity of their IPs in the various SHP ChIP-qPCR experiments presented in this manuscript, the authors should use chromatin isolated from the livers of Shp KO animals. Any enrichment detected in WT animals should be lost with Shp-KO mice and would satisfy the legitimate concern of any potential reader from the Shp field. The same precaution should be taken with the SHP ChIP assays using FGF19 treatment.

Response: In the revised manuscript, in addition to the IgG control data (Fig. 1c), we have added additional ChIP data for two key genes, *Pemt* and *Gnmt*, comparing C57BL/6 WT mice and SHP-KO mice, demonstrating that the SHP antibody is specific for SHP (Fig. 1d, below). In addition, detailed information of mouse liver ChIP experiment to detect SHP occupancy was added in the supplementary methods.

d SHP occupancy

Fig. 1. FGF19 inhibition of hepatic PC production is SHP-dependent. Effects of FGF19 treatment for 2 h on SHP occupancy determined by ChIP at 1C genes, which was confirmed at *Pemt* and *Gnmt* genes in C57BL/6 and SHP-KO mice. Means +/- SD (n=5 mice) are shown, and statistical significance was measured using the two-way ANOVA with the FDR post-test and *P<0.05, **P<0.01, and NS, not statistically significant.

3. The authors interchangeably use WT and C57BL6, and it is not clear whether the WT are littermates of the KO animals, or just C57BL6 mice. It is not preferable to use C57BL6 mice as controls for the various KO mice, because of various reasons, including differences in background, microbiota, etc. The gold standard is still to use littermates for every mutant line.

Response: Reviewer 1 had the same concern. Please see the response to comment 1 of reviewer 1.

4. Did the authors exclude the possibility that increased NR5A2 transcriptional activity may account for the increase in 1C cycle transcripts as was previously published (PMID 26267291). Although here the authors demonstrate low NR5A2 binding motifs within the SHP peaks, it could be sufficient to affect the expression of the 1C cycle genes, especially in the absence of its transcriptional repressor SHP. In addition, it would be informative if the sequence of binding motifs of all TFs in Fig 3A are listed in the manuscript.

Response: We have listed (Table S3) the sequences of the TF motifs summarized in Fig. 3a. In addition to AhR, other transcription factors, such as NR5A2 (LRH-1), could affect transactivation of 1C genes. Consistent with a recent study (Wagner et al., Hepatology, 2015), we confirmed in luciferase assays LRH-1, as well as AhR, transactivates the *Pemt* and *Gnmt* promoters (Fig. S5, below) and expression of SHP inhibits both, although effects of LRH-1 on *Pemt* were modest. In the revision manuscript, we have modified the result and discussion parts with regard to LRH-1.

Figure S5. Hepa1c1c7 cells were transfected with a *Pemt-luc* or *Gnmt-luc* construct containing the WT AhR binding site in the SHP binding region of the *Pemt* promoter along with expression plasmids as indicated. After 2 days, luciferase activities were measured. Means +/- SD are shown (n=5), and statistical significance was measured using one-way ANOVA with the FDR post-test. *P<0.05, **P<0.01, and NS, statistically not significant.

5. In Fig. 7A mice were either injected with an adenovirus encoding GFP alone or with a double adenovirus injection (Ad-CA-Ahr + Ad-SHP WT). Does this mean that these mice received double the number of viral particles?

Response: The same number of viral particles were injected in each case. For the Ad-CA-AhR only sample, the number of particles was adjusted by adding Ad-GFP control virus. This has now been noted in the Fig. 7 legend.

6. The authors show in the last figure that PC and PE levels are dramatically downregulated, but PC/PE ratio increased in severe NAFLD (NASH) patients. The relative levels of PC and PE being so low, how would an increase in this ratio be relevant?

Response: We agree that because of the extremely low levels, the PC/PE ratio is not really relevant, even though the trend of the ratio is in the expected direction. We have retained the data for completeness, but have modified the discussion of this result.

Minor comments:

1. In the graphs, Fs and Fd are used without indication of what it means.

Response: We have defined Fs and Fd in the legends.

Reviewer #3 (Remarks to the Author):

This manuscript is well written and is on a timely and important topic. The data is nicely presented and mostly supports the stated conclusions that significantly advances this field of research. However, there are a number of issues that need to be addressed to have the data firmly support the conclusions.

1. In figure 4b the authors have used PD98059 as an inhibitor of MAP kinase pathway and there appears to be a partial decrease in nuclear AhR levels, this can be explained by the fact that PD98059 is an AhR antagonist (Reiners, JJ, et al. 1998. Mol Pharmacol, 53, 438-45), this information needs to be added to the manuscript.

Response: We added the information in the manuscript as suggested.

2. In figure 4b the authors have shown that the AhR is retained in the nucleus after insulin addition in Hepa1c1c7 cells. However, whether the AHR is heterodimerized with ARNT has not been established. Since the ability of insulin to activate the AhR is a key point (e.g. Fig. 8d) the author need to illustrate that AHR/ARNT heterodimer is actually formed by performing a gel shift analysis. Also, insulin cause Cyp1a1 activation in the Hepa1c1c7 cells should also be included.

Response: We thank the reviewer for raising this important issue of the AHR/ARNT heterodimer. We believe the ChIP assay provides a more relevant way to analyze protein/DNA interaction in an in vivo chromatin context. Thus, in the revision work, 1) first, we performed re-ChIP assay to examine the co-occupancy of AhR and ARNT (Fig. 4f, below) at the *Pemt* promoter. In the re-ChIP assays, feeding (Fig. 4f) or insulin (new Fig. S4, below) increased ARNT occupancy in AhR-bound chromatin, indicating that both AhR and ARNT are present at the *Pemt* promoter. 2) Second, we performed feeding-time course ChIP assays to determine the ARNT occupancy (Fig. 4h, below). Remarkably, the time-course binding of ARNT was very similar to that of AhR (Fig. 4h). In addition, we have examined the effect of insulin on *Cyp1a1* expression and showed that *Cyp1a1*, like *Pemt*, pre-mRNA levels were increased by insulin treatment (Fig. 4c, below).

Fig. 4. SHP inhibits AhR transactivation of *Pemt* in response to feeding or FGF19 treatment. (c) Hepa1c1c7 cells were treated with insulin for 15 min or 45 min, and pre-mRNA levels of *Pemt* and *Cyp1a1* were measured (n=6). (f) Mice (n=5 mice/group) were refed for 6 h after fasting overnight. Liver chromatin was immunoprecipitated with AhR antibody, then eluted, and re-precipitated with ARNT or SHP antibody to examine the occupancy of SHP at AhR-bound chromatin at the *Pemt* promoter. (h) Mice (n=5 mice/group) were fasted and refed for the indicated times, and the occupancy of AhR and ARNT at the *Pemt* promoter was determined by liver ChIP assay. Means +/- SD are shown, and statistical significance was measured using the (c) one- or (f, h) two-way ANOVA with the FDR post-test. *P<0.05, **P<0.01, and NS, statistically not significant.

Figure S4. The chromatin isolated from Hepa1c1c7 cells treated with insulin for 1 h was immunoprecipitated with AhR antibody, then eluted, and re-precipitated with ARNT antibody to examine the occupancy of ARNT at AhR-bound chromatin at the *Pemt* promoter. Means +/- SD are shown (n=5), and statistical significance was measured using the two-way ANOVA with the FDR post-test. *P<0.05, **P<0.01, and NS, statistically not significant.

3. Comments about experiments in Figure 8:

A. The human AhR has a molecular weight of 105 kDa that is significantly higher than the mouse AhR (i.e. 95 kDa). Yet the western blots of the CA-AhR and the human AhR show the same molecular weight, the authors need to examine the molecular weight standards and how they were placed on the western blots. In addition, in the human AHR liver sample western blot data the authors need a positive control from a human cell line in ensure that the band they are detecting actually is the AhR.

Response: In the original Figure 8, the mobility of human appeared to be more than 100 K based on the markers, but this was incorrect. In the revision work, we examined mouse AhR in Hepa1c1c7 cells or exogenously expressed mouse CA-AhR and human AhR from HepG2 cells or the human liver samples (below). The AhR in the two human samples had the same mobility which was slightly slower than the mouse samples as expected. We have corrected the positions of the markers in Fig. 8 and added the control comparisons of mouse and human as Fig. S10. We thank the reviewer for pointing out this mistake.

Figure S10. Hepa1c1c7 cells were transfected with empty vector or the plasmids expressing CA-AhR. In addition, human HepG2 cells were also grown in normal media. After 2 days, protein was isolated from the cells of Hepa1c1c7 and HepG2 as well as human liver tissue, and immunoblotting was performed to measure the protein expression level.

B. The human AhR is very susceptible to proteolysis yet the western blot for the human liver samples does not show any degradation products. In the methods section, how the liver samples were prepared needs to be added to the methods, were protease inhibitors added.

Response: Detailed information about the preparation of liver extracts, including protease inhibitors and phosphatase inhibitors, has been added in the Supplemental Methods.

C. The mRNA data for the Ahr in the 15 human liver samples shows a wide range of variation in the steatosis samples, yet the 5 samples shown in the western blot were quite consistent. Did the authors select a specific subset of samples to analyze by western blot analysis?

Response: In IB analysis, to compare protein levels between experimental groups in one gel, three human samples of the 15 were randomly selected independent of the mRNA results and pooled to produce the 5 samples for analysis. Pooling of the sample would tend to reduce the variation.

4. More information about the various mouse models should be added to the methods section, are they all congenic on a C57BL6/J background?

Response: Reviewer 1 had the same concern. Please see the response to comments 1 and 2 of reviewer 1.

5. More details of the actual ChIP protocol should be included so that someone could actually repeat the experiment.

Response: As suggested by the reviewer, detailed information of ChIP experiment was added in the supplementary methods.

We thank the reviewers for their constructive comments. In response, we have included substantial new data and believe we have addressed all of the key issues that were raised. We believe the new data and other changes greatly strengthen the manuscript.

Reviewer #1 (Remarks to the Author):

The authors have satisfactorily answered to all the questions and comments raised by this reviewer. The manuscript adds a highly novel and interesting layer of complexity to the role of 1-carbon metabolism in liver biology and pathobiology.

Reviewer #2 (Remarks to the Author):

The authors have addressed all my major concerns and I have no further questions.

Reviewer #3 (Remarks to the Author):

The authors have meaningfully addressed the majority of the reviewer's comments, which has resulted in a significantly improved manuscript that now firmly supports the stated conclusions. However, there is one major concern that needs to be resolved.

1. The CA-AHR was not explained as to where the vector pCMX-CA-AhR was obtained from and what is its structure of the CA-AHR cDNA. I assume that the construct utilized is similar to what others have used which has amino acids residues 288-421 deleted. This would yield a truncated protein that would migrate as a smaller protein than the full-length WT mAHR. In a PNAS article (Andersson, P., 2002) the difference in the relative molecular weight of the CA-AHR and WT AHR is easily observed. Yet in figure 7b and in figure S10 the band shown for the CA-AHR and WT AHR have the exact same molecular weight on the western blot. Can the authors explain how this is possible?

Response to Reviewers

Reviewer #1 (Remarks to the Author):

The authors have satisfactorily answered to all the questions and comments raised by this reviewer. The manuscript adds a highly novel and interesting layer of complexity to the role of 1-carbon metabolism in liver biology and pathobiology.

We thank the reviewer.

Reviewer #2 (Remarks to the Author):

The authors have addressed all my major concerns and I have no further questions.

We thank the reviewer.

Reviewer #3

"The authors have meaningfully addressed the majority of the reviewer's comments, which has resulted in a significantly improved manuscript that now firmly supports the stated conclusions. However, there is one major concern that needs to be resolved.

Comments:

1. The CA-AHR was not explained as to where the vector pCMX-CA-AhR was obtained from and what is its structure of the CA-AHR cDNA. I assume that the construct utilized is similar to what others have used which has amino acids residues 288-421 deleted. This would yield a truncated protein that would migrate as a smaller protein than the full-length WT mAHR. In a PNAS article (Andersson, P., 2002) the difference in the relative molecular weight of the CA-AHR and WT AHR is easily observed. Yet in figure 7b and in figure S10 the band shown for the CA-AHR and WT AHR have the exact same molecular weight on the western blot. Can the authors explain how this is possible? "

Response:

We apologize for the confusion. The CA-AhR used in our study was the human version not the mouse. We obtained pCMX-CA-AhR for human AhR, which was constructed by deleting the human AHR region encoding the minimal ligand-binding domain (amino

acids 273 to 432), from Dr. Wen Xie's group (Lu et al, Hepatology. 2015 61:1908-19, 2015). This deletion of 159 amino acids reduces the MW by about 16 K so that the MW of human CA-AhR is similar to that of mouse WT AhR which is about 18K to 20K smaller than the human WT. The human CA-AhR was expressed in mouse Hepa1c1c7 cells (Fig. S10) and, as expected, migrates slower than the human WT AhR (Fig. S10, shown below) and similar to the mouse WT AhR (Fig. S10 and Fig. 7b).

We have added the source of the CA-AhR to the Methods section and noted that the mouse WT-AhR and the human CA-AhR have similar MWs. We also clarified that CA-AhR is the human form in the legends to Fig. 7b and Fig. S10.

Fig. S10